

# A crash-testing framework for predictive uncertainty assessment when forecasting high flows in an extrapolation context

Lionel Berthet[1], François Bourgin[2], Charles Perrin[3], Julie Viatgé[3], Renaud Marty[1], and Olivier Piotte[4]

[1]DREAL Centre-Val de Loire, Loire Cher & Indre Flood Forecasting Service, Orléans, France
[2]IFSTTAR, GERS, EE, Bouguenais, France
[3]IRSTEA, HYCAR Research Unit, Antony, France
[4]Ministry for the Ecological and Inclusive Transition, SCHAPI, Toulouse, France

**Correspondence:** lionel.berthet@developpement-durable.gouv.fr

**Abstract.** An increasing number of flood forecasting services assess and communicate the uncertainty associated with their forecasts. While obtaining reliable forecasts is a key issue, it is a challenging task, especially when forecasting high flows in an extrapolation context, i.e. when the event magnitude is larger than what was observed before. In this study, we present a crash-testing framework that evaluates the quality of hydrological forecasts in an extrapolation context. The experiment setup is

based on i) a large set of catchments in France, ii) the GRP rainfall-runoff model designed for flood forecasting and used by the French operational services and iii) an empirical hydrologic uncertainty processor designed to estimate conditional predictive uncertainty from the hydrological model residuals. The variants of the uncertainty processor used in this study differ in the data transformation they used (log, Box-Cox and log-sinh) to account for heteroscedasticity. Different data subsets were selected based on a preliminary event selection. Various aspects of the probabilistic performance of the variants of the hydrologic

uncertainty processor, reliability, sharpness and overall quality, were evaluated. Overall, the results highlight the challenge of uncertainty quantification when forecasting high flows. They show a significant drop in reliability when forecasting high flows in an extrapolation context and considerable variability among catchments and across lead times. The increase in statistical treatment complexity did not result in significant improvement, which suggests that a parsimonious and easily understandable data transformation such as the log transformation or the Box-Cox transformation with a parameter between 0.1 and 0.3 can

be a reasonable choice for flood forecasting.

## 1 Introduction

### 1.1 The big one: dream or nightmare for the forecaster?

In many countries, operational flood forecasting services (FFS) issue forecasts routinely throughout the year and during rare

or critical events. End-users are mostly concerned by the largest and most damaging floods, when critical decisions have to be



made. For such events, operational flood forecasters must get prepared to deal with extrapolation, i.e. to work on events of a magnitude that they and their models have seldom or never met before.

The relevance of simulation models and their calibration in evolving conditions, such as contrasted climate conditions and climate change has been studied by several authors. For example, Wilby (2005), Vaze et al. (2010), Merz et al. (2011) and

Brigode et al. (2013) explored the transferability of hydrological model parameters from one period to another and assessed the uncertainty associated with this parametrization, while Coron et al. (2012) proposed a generalisation of the differential split-sample test (Klemeš, 1986). In spite of its importance in operational contexts, only a few studies have addressed the extrapolation issue for flow forecasting, to the best of our knowledge, with the notable exception of data-driven approaches (e.g., Todini, 2007). Imrie et al. (2000), Cigizoglu (2003) and Giustolisi and Laucelli (2005) evaluated the ability of trained artificial

neural networks (ANNs) to extrapolate beyond the calibration data and showed that ANNs used for hydrological modelling may have poor generalisation properties. Singh et al. (2013) studied the impact of extrapolation on hydrological prediction with a conceptual model, and Barbetta et al. (2017) expressed concerns for the extrapolation context defined as floods of a magnitude not encountered during the calibration phase.

Addressing the extrapolation issue involves a number of methodological difficulties. Some data issues are specific to the data

used for hydrological modelling, such as the rating curve reliability (Lang et al., 2010). Other well-known issues are related to the calibration process: are the parameters, which are calibrated on a limited set of data, representative or at least somewhat adapted to other contexts? A robust modelling approach for operational flood forecasting, i.e. a method able to provide relevant forecasts in conditions not met during the calibration phase, requires paying special attention to the behaviour of hydrological models and the assessment of predictive uncertainty in an extrapolation context.

**1.2 Obtaining reliable forecasts remains a challenging task**

Even if significant progress has been made and implemented in operational systems (e.g., Bennett et al., 2014; Demargne et al., 2014; Pagano et al., 2014), some uncertainty remains. Communication of reliable predictive uncertainty information is therefore required to improve crisis management and decision making (Todini, 2004; Pappenberger and Beven, 2006; Demeritt et al., 2007; Verkade and Werner, 2011). Hereafter, reliability is defined as the statistical consistency between the observations

and the predictive distributions (Gneiting et al., 2007).

The uncertainty associated with operational forecasts is most often described by a predictive uncertainty distribution. Assessing a reliable predictive uncertainty distribution is challenging because hydrological forecasts yield residuals that show heteroscedasticity, i.e. an increase in the uncertainty variance with discharge, time auto-correlation, skewness, etc. Some studies (e.g., Yang et al., 2007; Schoups and Vrugt, 2010) account for these properties for the calibration of hydrological models

within a Bayesian framework, using specific formulations of likelihood. In an extrapolation context, it is of utter importance that the predictive uncertainty assessment provides a correct description of the heteroscedasticity, explicitly or implicitly.

Various approaches for uncertainty assessment have been developed to assess the uncertainty in hydrological predictions (see e.g., Montanari, 2011). The first step consists in identifying the different sources of uncertainty or at least the most important ones that have to be taken into account given a specific context. In the context of flood forecasting, decomposing the total





uncertainty into its two main components is now common: the input uncertainty (mainly the meteorological forecast uncertainty) and the modelling uncertainty, as proposed by Krzysztofowicz (1999). More generally, the predictive uncertainty due to various sources may be explicitly modelled and propagated through the modelling chain, while the "remaining" uncertainty (from the other sources) may then be assessed by statistical post-processing.

### 1.2.1 Modelling each source of uncertainty

A first approach intends to model each source of uncertainty separately and to propagate these uncertainties through the modelling chain (Renard et al., 2010). The heteroscedasticity of the predictive uncertainty distribution results from the separate modelling of each source of uncertainty and from the statistical model specification. While this approach is promising, operational application can be hindered by the challenge of making the hydrological modelling uncertainty explicit, as pointed out by Salamon and Feyen (2009).

In particular, the ensemble approaches intend to account for meteorological forecast uncertainty. They are increasingly popular in the research and the operational forecasting communities. An increasing number of hydrological ensemble forecasting systems are in operational use and have proved their usefulness, e.g. the European Flood Awareness System (EFAS: Ramos et al., 2007; Thielen et al., 2009; Pappenberger et al., 2011, 2016) and the Hydrologic Ensemble Forecast Service (HEFS: e.g. Demargne et al., 2014).

Multi-model approaches can be used to assess modelling uncertainty (Velazquez et al., 2010; Seiller et al., 2017). While promising, this approach requires the implementation and the maintenance of a large number of models, which can be burdensome in operational conditions. There is no evidence that such an approach ensures that the heteroscedasticity of the predictive uncertainty distribution would be correctly assessed.

In forecasting mode, data assimilation schemes based on statistical modelling are of common use to reduce and assess the predictive uncertainty. Some algorithms such as particle filters (Moradkhani et al., 2005a; Salamon and Feyen, 2009; Abbaszadeh et al., 2018) or the ensemble Kalman filter (Moradkhani et al., 2005b) provide an assessment of the predictive uncertainty as a direct result of data assimilation ("in the loop"). Some of these approaches can explicitly account for the desired properties of the predictive uncertainty distribution, such as heteroscedasticity, through the likelihood formulation.

### 1.2.2 Post-processing approaches

Alternatively, numerous post-processors of deterministic or probabilistic models have been developed to account for the uncertainty from sources that are not modelled explicitly. They differ in several aspects (see a recent review by Li et al., 2017). Most approaches are conditional: the predictive uncertainty is modelled with respect to a predictor, which most often is the forecasted value (Todini, 2007, 2009). Some methods are based on predictive distribution modelling, while others can be described as "distribution-free", as mentioned by Breiman (2001). Among the former, many approaches are built in a statistical regression framework to assess the total or remaining predictive uncertainty. Examples are the Hydrologic Uncertainty Processor (HUP) in a Bayesian forecasting system (BFS) framework (Krzysztofowicz, 1999; Krzysztofowicz and Maranzano, 2004), the Model Conditional Processor (MCP: Todini, 2008; Coccia and Todini, 2011; Barbetta et al., 2017), the meta-Gaussian model



of Montanari and Grossi (2008) or the Bayesian joint probability (BJP) method (Wang et al., 2009), among others. The latter approaches build a description of the predictive residuals from past error series, such as data learning-algorithms (Solomatine and Shrestha, 2009). Some related methods are the non-parametric approach of Van Steenbergen et al. (2012), the empirical hydrological uncertainty processor of Bourgin et al. (2014) or the k-nearest neighbours method of Wani et al. (2017). The

Quantile Regression (QR) framework (Weerts et al., 2011; Dogulu et al., 2015; Verkade et al., 2017) lies in between in that it introduces an assumption of a linear relationship between the forecasted discharge and the quantiles of interest.

These approaches are not exclusive of each other. Even when future precipitation is the main source of uncertainty, post-processing is often required to produce reliable hydrological ensembles (Zalachori et al., 2012; Hemri et al., 2015; Abaza et al., 2017; Sharma et al., 2018). Thus, many operational flood forecasting services use post-processing techniques to assess hydro-

logical modelling uncertainty, while meteorological uncertainty is taken into account separately (Berthet and Piotte, 2014). Post-processors are then trained with "perfect" future rainfall (i.e., equal to the observations). Moreover, even for assessing modelling uncertainty, using several methodologies together may allow one to combine their respective strengths.

Note that many of these approaches use a variable transformation to handle the heteroscedasticity. Some are non-parametric, while others use a few parameters, allowing more flexibility in the predictive distribution assessment. More details on com-

monly used variable transformations are presented in Section 2.1.4.

## 1.3 Scope

In this article, we focus on uncertainty assessment with a post-processing approach based on residuals modelling. Del Giudice et al. (2013) and McInerney et al. (2017) presented interesting comparisons of different variable transformations used for residuals modelling. Yet, their studies do not focus on the extrapolation context. Since it is not possible to achieve reliable predictive

uncertainty assessment in an extrapolation context if heteroscedasticity is not properly taken into account, the objectives of this article are:

- to present a framework aimed at testing the hydrological modelling and uncertainty assessment in the extrapolation context;

- to assess the ability and the robustness of a post-processor to provide reliable predictive uncertainty assessment for large

floods when different variable transformations are used;

- to provide guidance for operational flood forecasting system development.

We attempt to answer two questions: (a) Can we improve residuals modelling with an adequate variable transformation in an extrapolation context? (b) Do more flexible transformations, such as the log-sinh transformation, help in obtaining more reliable predictive uncertainty assessment?

Section 2 describes the data, the forecast model, the post-processor and the testing methodology chosen to address these questions. Section 3 presents the results of the numerical experiments that are then discussed in Sections 4 and 5. Finally, a number of conclusions and perspectives are proposed.





**Table 1.** Characteristics of the 154 catchments, computed over the 1997 – 2006 data series.

| | Percentiles | | | | | | |
|---|---|---|---|---|---|---|---|
| | 0 | 0.05 | 0.25 | 0.50 | 0.75 | 0.95 | 1 |
| Catchment area (km$^2$) | 9 | 27 | 79 | 184 | 399 | 942 | 3,260 |
| Average altitude (m above sea level) | 64 | 92 | 188 | 376 | 589 | 897 | 1,050 |
| Average slope (%) | 2 | 3 | 6 | 9 | 18 | 32 | 39 |
| Lag time (h) | 3 | 5 | 9 | 12 | 19 | 29 | 33 |
| Mean annual rainfall (mm/yr) | 639 | 727 | 876 | 1,003 | 1,230 | 1,501 | 1,841 |
| Mean annual potential evapotranspiration (mm/yr) | 549 | 549 | 631 | 659 | 700 | 772 | 722 |
| Mean annual discharge (mm/yr) | 53 | 142 | 262 | 394 | 583 | 1,114 | 1,663 |
| Mean annual discharge (m$^3$/s) | 1 | 1 | 1 | 2 | 5 | 17 | 53 |
| Maximum hourly rainfall (mm/h) | 10 | 12 | 16 | 20 | 26 | 41 | 61 |
| Quantile 0.99 of the hourly discharge (m$^3$/s) | 1 | 2 | 6 | 15 | 33 | 115 | 296 |

## 2 Data and methods

### 2.1 Data and forecasting model

#### 2.1.1 Catchments and hydrological data

We used a set of 154 unregulated catchments spread throughout France (Fig. 1) to test our hypotheses over various hydrological
5   regimes and forecasting contexts (Andréassian et al., 2006; Gupta et al., 2014). They represent a large variability in climate,
topography and geology in France (Table 1), although their hydrological regimes are little or not at all influenced by snow
accumulation. Hourly rainfall, potential evapotranspiration (PE) and streamflow data series were available over the 1997 –
2006 period. PE was estimated using a temperature-based formula (Oudin et al., 2005). Rainfall and temperature data come
from a radar-based reanalysis produced by Météo-France (Tabary et al., 2012). Discharge data were extracted from the national
10  streamflow HYDRO archive (www.hydro.eaufrance.fr). For each catchment, the lag time LT is estimated as the lag time
maximising the cross-correlation between rainfall and discharge time series.

#### 2.1.2 Hydrological model

We used discharge forecasts computed by the GRP rainfall-runoff model. The GRP model is designed for flood forecasting
and is currently used by the Flood Forecasting Services in France in operational conditions (Furusho et al., 2016; Viatgé et al.,
15  2018b). It is a deterministic lumped storage-type model that uses catchment areal rainfall and PE as inputs. The model also
assimilates discharge observations available when issuing a forecast to update the main state variable of the routing function





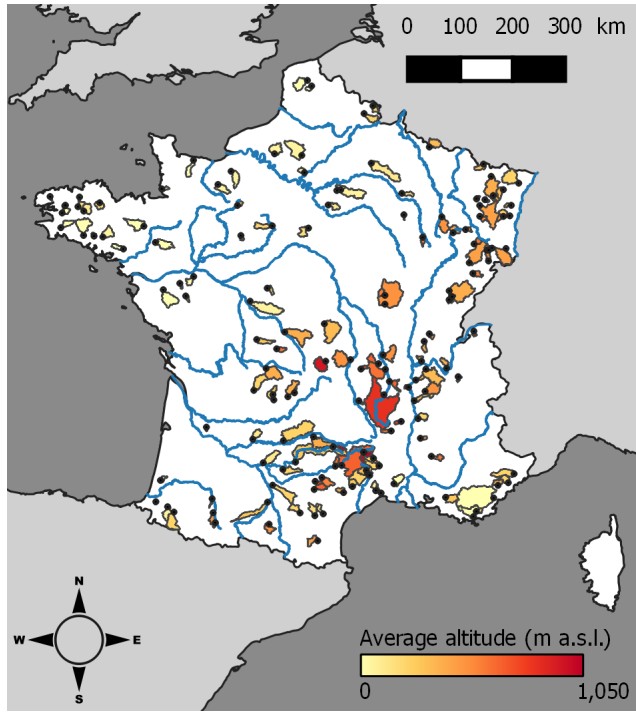

**Figure 1.** The set of 154 unregulated catchments used in this study. Average altitude is given in meters above sea level ($m$ a.s.l.).

and to update the output discharge. In this study, it is run at the hourly time step and forecasts are issued for several lead times ranging from 1 to 72 h. More details about the GRP model can be found in Appendix A.

Since herein only the ability of the post-processor to extrapolate uncertainty quantification is studied, the model is calibrated in forecasting mode over the 10-year series by minimising the sum of squared errors for a lead time taken as the lag time LT.

5   The results will be presented for four lead times, LT / 2, LT , 2 LT and 3 LT, to cover the different behaviours that can be seen when data assimilation is used to reduce errors in an operational flood forecasting context.

### 2.1.3   Empirical hydrological uncertainty processor (EHUP)

We used the empirical hydrological uncertainty processor (EHUP) presented in Bourgin et al. (2014). It is a data-based and non-parametric approach to estimate the conditional predictive uncertainty. This post-processor was compared to other post-

10   processors in earlier studies and proved to provide relevant results (Bourgin, 2014). It is now used by operational FFS in France under the operational tool called OTAMIN (Viatgé et al., 2018a). Separately for each lead-time, empirical quantiles of errors are estimated for different flow groups obtained by ordering the forecast-observation pairs according to the forecasted values. Since we focus here on the extrapolation case, only the highest flow group containing the top 5% pairs ranked by simulated values is used. This threshold is chosen as a compromise between focusing at the highest values and using a sufficiently large





number of forecast-observation pairs when estimating empirical quantiles of errors. Heteroscedasticity of the extrapolated predictive distributions is taken into account using different data transformations described in the next subsection.

### 2.1.4 The different transformation families

Many uncertainty assessment methods mentioned in the introduction use a variable transformation to handle the heteroscedasticity of the residuals. Here, we briefly recall a number of variable transformations commonly used in hydrological modelling. Let $y$ and $\tilde{y}$ be the observed and forecasted variables (here, the discharge) and $\varepsilon = y - \tilde{y}$ the residuals. When using a transformation $g$, we consider the residuals $\varepsilon' = g(y) - g(\tilde{y})$.

The normal quantile transformation (NQT) is a non-parametric transformation linking a given distribution and the Gaussian distribution, quantile by quantile:

$$g_1(y) = \text{NQT}(y) = \phi^{-1}(F(y))$$

where $\phi$ is the normal Gaussian cumulative density function (cdf) and $F$ is the cdf of a variable $Y$. When used in a post-processor, $F$ is most often empirically computed from a large sample of residuals. While several hydrological processors such as the HUP, MCP and the Quantile Regression encompass the NQT-transformed variables, Bogner et al. (2012) warn against the drawbacks of this transformation, which is by construction not suited for the extrapolation context and requires additional assumptions to model the tails of the distribution (Weerts et al., 2011; Coccia and Todini, 2011).

The next three transformations are analytical. The log transformation is commonly used (e.g., Morawietz et al., 2011):

$$g_2(y) = \log(y + a)$$

where $a$ is a small positive constant to deal with y values close to 0. It can be taken equal to 0 when focusing on large discharge values. This transformation is then non-parametric. Applying a statistical model on residuals computed on log-transformed variables may be interpreted as using a corresponding model on multiplicative error (e.g., assuming a Gaussian model for residuals of log-transformed discharges is equivalent to a log-Normal model on the multiplicative errors $y/\tilde{y}$). Therefore, it may be adapted to strongly heretoscedastic behaviours. It has been used successfully to assess hydrological uncertainty (Yang et al., 2007; Schoups and Vrugt, 2010; Bourgin et al., 2014).

The Box-Cox transformation (Box and Cox, 1964) is a classic one-parameter transformation that is quite popular in the hydrological community (e.g., Yang et al., 2007; Wang et al., 2009; Hemri et al., 2015; Reichert and Mieleitner, 2009; Singh et al., 2013; Del Giudice et al., 2013):

$$g_3(y) = \frac{(y + a)^{\lambda} - 1}{\lambda} \quad \text{if} \quad \lambda \neq 0 \quad \text{and} \quad g_3(y) = g_2(y) \quad \text{if} \quad \lambda = 0$$

The Box-Cox transformation makes it possible to cover very different behaviours. The log transformation is a special case of the Box-Cox transformation when the calibration results in $\lambda = 0$. In contrast, applying the Box-Cox transformation with $\lambda = 1$ on the variable $y$ to model the distribution of their residuals is equivalent to applying no transformation (Fig. 2). McInerney et al. (2017) obtained their best results with $\lambda = 0.2$ over 17 perennial catchments.





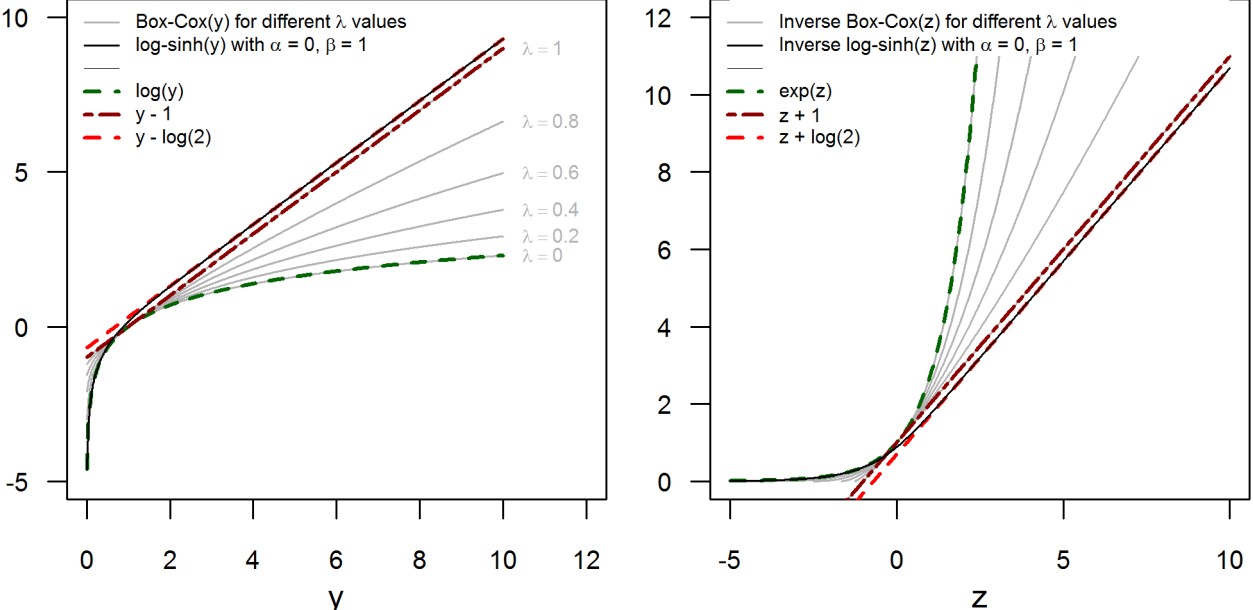

**Figure 2.** Left panel: the Box-Cox transformation for various parameter values (grey straight lines) and the log-sinh transformation for one parametrization (black straight line). Right panel: the corresponding inverse transformations. The Box-Cox transformation provides different behaviours, depending on its parameter value ($\lambda$). It ranges from an affine transformation (equivalent to no transformation; red dashed line) to the log transformation (thick green straight line). With a single parametrization, the log-sinh transformation can be equivalent to the log transformation for values of $y$ much smaller than the value of its parameter $\beta$ and equivalent to an affine transformation for large values of $y$ (much higher than $\beta$).

More recently, the log-sinh transformation has been proposed (Wang et al., 2012; Pagano et al., 2013). It is a two-parameter transformation:

$$g_{\alpha,\beta} : y \mapsto \beta \cdot \log\left(\sinh\left(\frac{\alpha + y}{\beta}\right)\right)$$

This transformation provides more flexibility. Indeed, for $y \gg \alpha$ and $y \gg \beta$, the log-sinh transformation reduces to no trans-
5 formation, while for $\alpha \ll y \ll \beta$, it is equivalent to the log transformation (Fig. 2). Thus, with the same parametrization, it can result in very different behaviours depending on the magnitude of the discharge. Applying no transformation may be intuitively attractive to model the residuals distribution for very large discharge values, when the variance is not longer expected to increase (homoscedastic behaviour). It is then particularly attractive when modelling predictive uncertainty in an extrapolation context, in order to avoid an excessively "explosive" assessment of the predictive uncertainty for large discharge values.
10 In addition to the log transformation used by Bourgin et al. (2014), in this study we tested the Box-Cox and the log-sinh transformations to explore more flexible ways to deal with the challenge of extrapolating prediction uncertainty distributions (Fig. 2). The impacts of the data transformations used in this study are illustrated in Fig. 3.





**Figure 3.** Predictive 0.1 and 0.9 quantiles when assessed with no transformation, the log-transformation, the Box-Cox transformation with its $\lambda$ parameter equal to 0.5 and the log-sinh with its parameters equal to 0.1 and 8, on the Ill River at Didenheim (668 km$^2$) for lead time LT: (a) as a function of the deterministic forecasted discharge and (b) for the flood event on 3 March 2006. The heteroscedasticity strongly differs from one variable transformation to another.

## 2.2 Methodology: a testing framework designed for extrapolation context assessment

### 2.2.1 Testing framework

The EHUP post-processor is a non-parametric approach based on the characteristics of the residuals distribution over a training data set. Moreover, the Box-Cox and the log-sinh transformations are parametric and require a calibration step. Therefore, the





**Table 2.** Characteristics of the events selected for the lead time LT over the 1997 – 2006 data series.

| | Percentiles | | | | | | |
| --- | --- | --- | --- | --- | --- | --- | --- |
| | 0 | 0.05 | 0.25 | 0.50 | 0.75 | 0.95 | 1 |
| Total length of events (days) | 663 | 1178 | 1299 | 1505 | 1808 | 2061 | 2762 |
| Number of events G1 | 28 | 65 | 114 | 141 | 193 | 276 | 434 |
| Number of events G2 | 8 | 16 | 23 | 32 | 43 | 58 | 170 |
| Number of events G3 | 7 | 12 | 19 | 26 | 35 | 46 | 162 |
| Median value of the peak discharges G1 ($m^3$/s) | 1 | 1 | 1 | 3 | 7 | 11 | 87 |
| Median value of the peak discharges G2 ($m^3$/s) | 1 | 2 | 5 | 11 | 24 | 49 | 254 |
| Median value of the peak discharges G3 ($m^3$/s) | 2 | 3 | 9 | 23 | 53 | 103 | 502 |

methodology adopted for this study is a split-sample scheme test inspired by the differential split-sample scheme of Klemeš (1986) and based on three data subsets: a data set for training the EHUP, a data set for calibrating the parameters of the variable transformation, and a control data set for evaluating the predictive distributions when extrapolating high flows.

### 2.2.2 Events selection

To populate the three data subsets with independent data, separate flood events were first selected by an iterative procedure similar to those presented in Lobligeois et al. (2014) and Ficchì et al. (2016): (1) the maximum discharge of the time series was selected, (2) within a 20-day period before (after) the peak flow, the beginning (end) of the event was placed at the first time step at which the streamflow is lower than 20% (25%) of the peak flow value, (3) the event was kept if there was less than 10% missing values, if the beginning and end of the event were lower than 66% of the peak flow value and if the peak value was

higher than 50% of the highest discharge value of the time series. A minimum time lapse of 24 h was enforced between two events, ensuring that consecutive events are not overlapping and that the autocorrelation between the time steps of two separate events remains limited.

The number of events and their characteristics vary greatly among catchments, as summarised in Table 2. Note that the events selected for one catchment can slightly differ for the four different lead times considered in this study, because the

selection was made using the forecasted discharge and not the observed discharge.

### 2.2.3 Selection of the data subsets

The selected events were then gathered into three events sets, G1, G2 and G3, based on the magnitude of their peaks and the number of useful time steps for each test phase (training of the EHUP post-processor, calibration of the variable transformations, evaluation of the predictive distributions): G1 contains the lowest events while the highest events are in G3.

The selection of the data subsets was tailored to study the behaviour of the post-processing approach in an extrapolation context. The control data subset had to encompass only time steps with simulated discharge values higher than those met





during the training and calibration steps. Similarly, the calibration data subset had to encompass time steps with simulated discharge values higher than those of the training subset.

To achieve these goals, only the time steps within flood events were used. We distinguished four data subsets, as illustrated in Fig. 4. The subset D1 gathered all the time steps of the events of the G1 group. Then the set D2 of the time steps of the events of the G2 group was split into two subsets: $D2_{sup}$ gathered all the time steps with forecasted discharge values higher than the maximum met on D1, and $D2_{inf}$ was filled with the other time steps. Finally, D3 was similarly filled with all the time steps of the G3 events with forecasted discharge values higher than the maximum met on D2.

The discharge thresholds used to populate the D1, $D2_{sup}$ and D3 subsets from the events belonging to the G1, G2 and G3 groups were chosen to ensure a sufficient number of time steps in every subset. We chose to set the minimum number of time steps in D3 and $D2_{sup}$ to 720 as a compromise between having enough data to evaluate the methods and keeping the extrapolation range sufficiently large. We lowered this limit to 500 for D1, since this subset was only used to build the empirical distribution by estimating 99 percentiles during the training step.

### 2.2.4 Calibration and evaluation steps

Since there are only one parameter for the Box-Cox transformation and two parameters for the log-sinh transformation, a simple calibration approach of the transformation parameters was chosen: the parameter space was explored by testing several parameter set values. For the Box-Cox transformation, 17 values for the $\lambda$ parameter were tested: from 0 to 1 with a step of 0.1 and with a refined mesh for the sub-intervals $0 - 0.1$ and $0.9 - 1$. For the log-sinh transformation, a grid of 200 $(\alpha, \beta)$ values was designed for each catchment based on the maximum value of the forecasted discharge on the D2 subset, as explained in greater detail in the Appendix B.

Note that the hydrological model was calibrated over the whole set of data (1997 – 2006) to make the best use of the data set, since this study focuses on the effect of extrapolation on the predictive uncertainty assessment only.

We used a two-step procedure, as illustrated in Fig. 5. In the first step, for each transformation and for each parameter set, the empirical residuals were computed over D1 and the calibration criterion was computed on $D2_{sup}$. The parameter set obtaining the best criterion value was selected. In the second step, the empirical residuals distribution was estimated on a data set which encompassed D1, $D2_{inf}$ and $D2_{sup}$ using the parameter set obtained during the calibration step, and the predictive uncertainty distribution was evaluated on the control data set D3. This second estimation of the residuals distribution made it possible to test the model from small to large degrees of extrapolation, on every catchment (see the discussion in Sect. 4.2).

### 2.3 Performance criteria and calibration

### 2.3.1 Probabilistic evaluation framework

Reliability was first assessed by a visual inspection of the Probability integral transform (PIT) diagrams (Laio and Tamea, 2007; Renard et al., 2010). Since this study was carried out over a large sample of catchments, two standard numerical criteria were used to summarise the results: the $\alpha$-index, which is directly related to the PIT diagram (Renard et al., 2010), and the





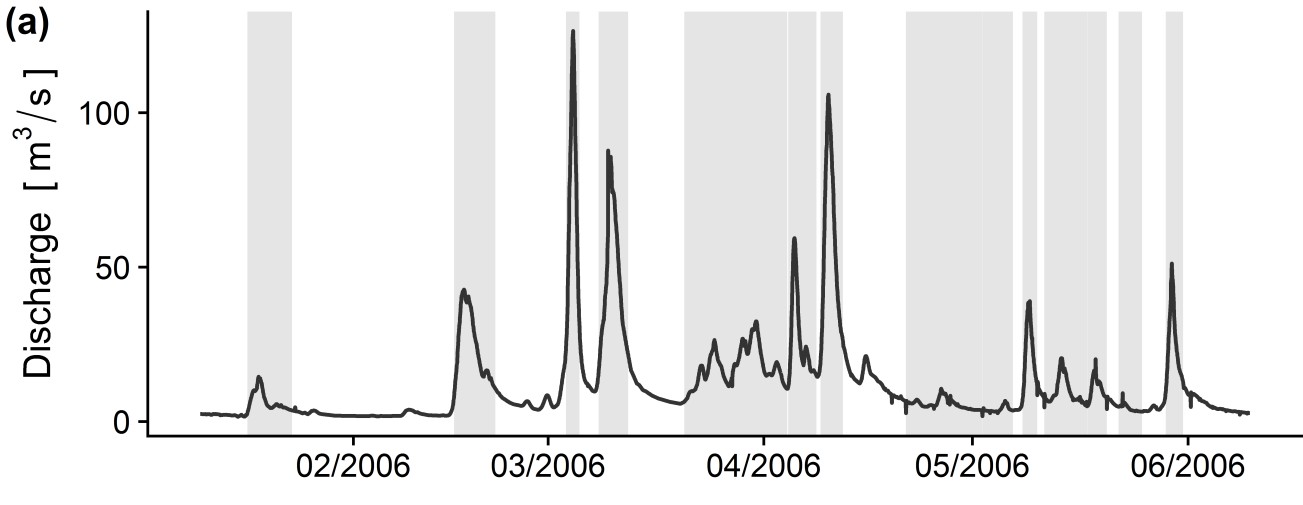

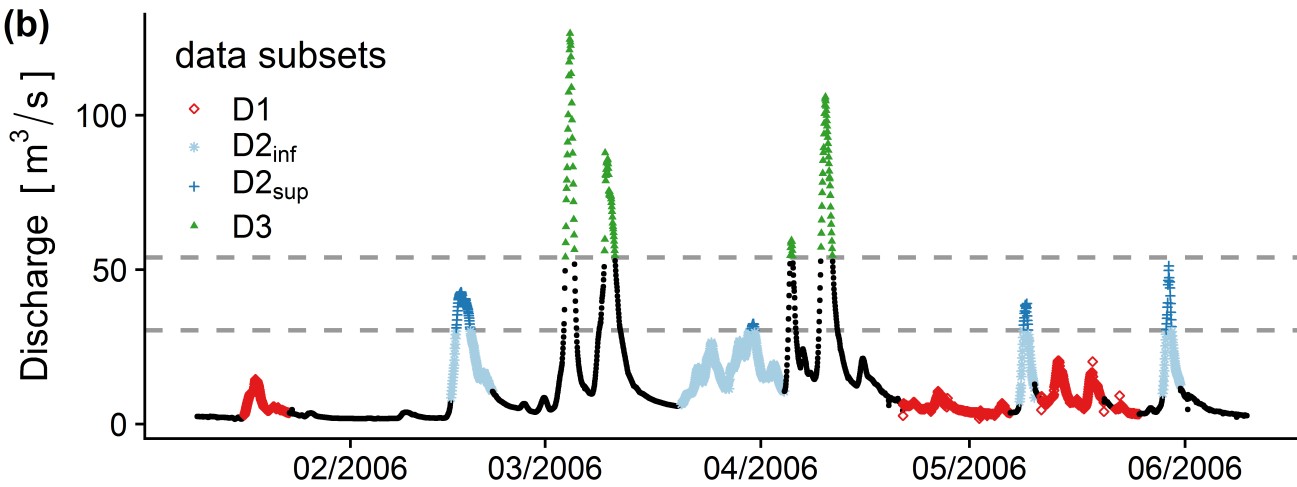

**Figure 4.** Illustration of (a) the selection of events in grey and (b) the selection of the four data subsets for the Ill River at Didenheim (668 $km^2$). The horizontal dashed lines show the thresholds between data subsets.

coverage rate of the 80% predictive intervals, used by the French operational FFS. The $\alpha$-index is equal to $1 - 2 \cdot A$, where $A$ is the area between the PIT curve and the bisector, and its value ranges from 0 to 1 (perfect reliability).




The overall quality of the probabilistic forecasts was evaluated with the Continuous Rank Probability Score (CRPS, Hersbach, 2000), which compares the predictive distribution to the observation:

$$\text{CRPS} = \frac{1}{N} \sum_{k=1}^{N} \int_{0}^{\infty} \left[ F_k\left(Q\right) - H\left(Q - Q_{k,obs}\right) \right]^2 dQ$$

where $F(Q)$ is the predictive cumulative distribution, $H$ the Heaviside function and $Q_{obs}$ is the observed value. We used a skill

score (CRPSS) to compare the mean CRPS to a reference, here the mean CRPS obtained from the unconditional climatology, i.e. from the distribution of the observed discharges over the events selected.

For operational purposes, the sharpness of the probabilistic forecasts was checked by measuring the mean width of the 80% predictive intervals. A non-dimensional relative sharpness index was obtained by dividing the mean width by the mean runoff:

$$1 - \frac{\overline{q_{90}\left(Q\right) - q_{10}\left(Q\right)}}{Q_{obs}}$$

where $q_{90}\left(Q\right)$ and $q_{10}\left(Q\right)$ are the upper and the lower bound of the 80% predictive interval for each forecast.

Finally, the accuracy of the forecasts was assessed using the Nash-Sutcliffe Efficiency (NSE) calculated with the mean values of the predictive distributions.

### 2.3.2 The calibration criterion

Since the calibration step aims at selecting the most reliable description of the residuals in extrapolation, the $\alpha$-index was used

to select the parameter set that yields the highest reliability for each catchment, each lead time and each transformation. While other choices were possible, we followed the paradigm presented by Gneiting et al. (2007): reliability has to be ensured before sharpness. Note that the CRPS could have been chosen, since it can be decomposed as the sum of two terms: reliability and sharpness (Hersbach, 2000). However, in the authors' experience, the latter is the controlling factor (Bourgin et al., 2014). Moreover, the CRPS values were often quite insensitive to the values of the log-sinh transformation parameters.

In cases where an equal value of the $\alpha$-index was obtained, we selected the parameter set that gave the best sharpness index. For the log-sinh transformation, there were still a few cases where an equal value of the sharpness index was obtained, revealing the lack of sensitivity of the transformation in some areas of the parameter space. For those cases, we chose to keep the parameter set that had the lowest $\gamma_1$ value and the $\gamma_2$ value closest to 1.





**Figure 5.** Residuals as a function of the forecast discharges in the transformed space. The horizontal dashed lines represent the 0.1, 0.25, 0.5, 0.75 and 0.9 quantiles of the residuals computed during the training phase of the EHUP post-processor, while the straight lines represent their use to assess the predictive uncertainty in extrapolation during (a) the calibration step of the variable transformation parameters and (b) the evaluation step of the predictive uncertainty. The vertical dashed lines show the beginning of the extrapolation range. Illustration from the Ill River at Didenheim, 668 km$^2$. Data used for the EHUP training at each step is sketched by a thick rectangle.



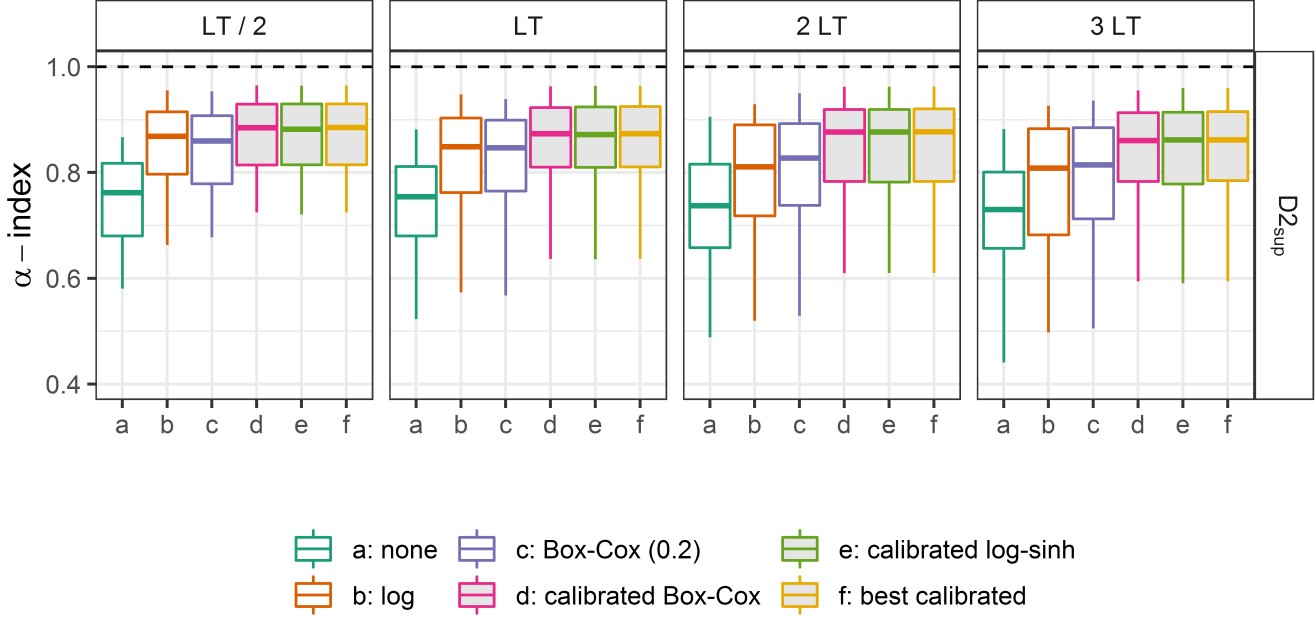

**Figure 6.** Distributions of the $\alpha$-index values on the calibration data set D2$_{sup}$, obtained with different transformations for four lead times (the filled boxplots represent the calibrated distributions). Boxplots (5th, 25th, 50th, 75th and 95th percentiles) synthesise the variety of scores over the catchments of the data set.

## 3   Results

### 3.1   Results on the calibration data set D2$_{sup}$

Figure 6 shows the distributions of the $\alpha$-index values obtained with different transformations on the calibration data set (D2$_{sup}$) for lead times LT / 2, LT, 2 LT, 3 LT. The distributions are summarised with boxplots. Clearly, not using any transformation leads to poorer reliability than any tested transformation. In addition, we note that the calibrated transformations provide better results (although not perfect) than the non-parametric ones on the calibration data set, as expected, and that there is no significant difference between the calibrated Box-Cox transformation (d), the calibrated log-sinh transformation (e) and the best performing transformation (f). Nevertheless, the uncalibrated log transformation and Box-Cox transformation with parameter $\lambda$ set at 0.2 ($BC_{\lambda=0.2}$) reach quite reliable forecasts. Interestingly, the log transformation provides the best results for the other criteria (not used as the objective function). Comparing the results obtained for the different lead times reveals that less reliable predictive distributions are obtained for longer lead times, in particular for the non-parametric transformations.

Figures 7 and 8 show the distribution of parameter values obtained for the Box-Cox and the log-sinh transformation during the calibration step. The distributions vary with lead time. While the log-transformation behaviour is frequently chosen for LT/2 and LT, the additive behaviour becomes more frequent for 2 LT and 3 LT. A similar conclusion can be drawn for the





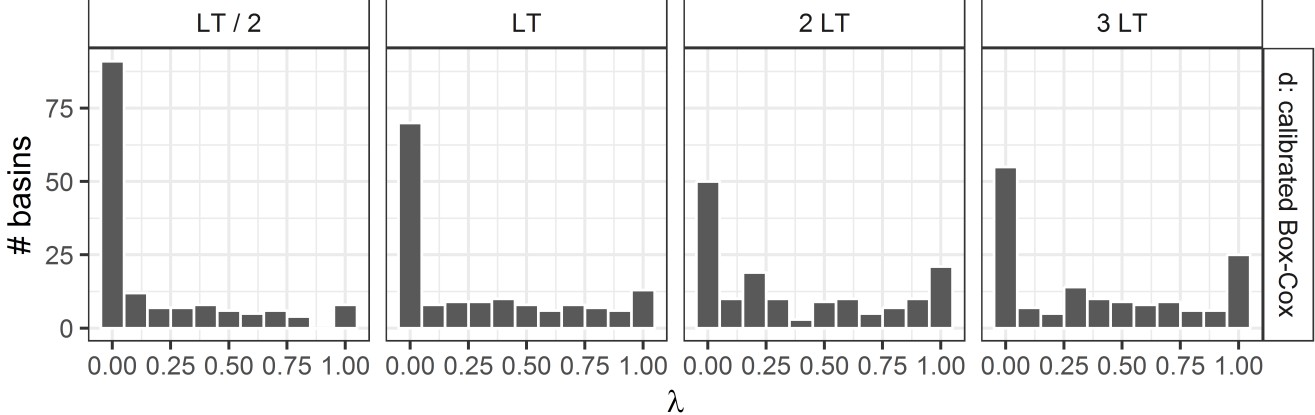

**Figure 7.** Distribution over the basins of the values of the Box-Cox transformation parameter obtained during the calibration step for the four different lead times.

log-sinh transformation: a low value of $\gamma_1$ and a high value of $\gamma_2$ yield a multiplicative behaviour that is frequently chosen, for all lead times, but less for 2 LT and 3 LT than for LT / 2 and LT. This explains in particular the loss of reliability that can be seen for the log transformation for LT 3 in Fig. 6. These results reveal that the extrapolation behaviour of the residuals distributions is complex. It varies among catchments and with lead time because of the strong impact of data assimilation.

### 3.2 Results on the D3 control data set

#### 3.2.1 Reliability

First, we conducted a visual inspection of the PIT diagrams, which convey an evaluation of the overall reliability of the probabilistic forecasts. They show quite different patterns on the set of catchments, highlighting bias or under-dispersion problems for some of them, as illustrated in Fig. 9.

Then the distribution of the $\alpha$-index values in Fig. 10 reveals a significant loss of reliability compared to the values obtained on the calibration data set (Fig. 6). We note that the log transformation is the most reliable approach for LT / 2 and is comparable to the Box-Cox transformation ($BC_{\lambda=0.2}$) for LT. With increasing lead time, the $BC_{\lambda=0.2}$ transformation becomes slightly better than the other transformations, including the calibrated ones. In addition, comparing the results obtained for the different lead times confirms that it is challenging to produce reliable predictive distributions when extrapolating at longer lead times. Overall, it means that the added value of the flexibility brought by the calibrated transformations is not transferable in an independent extrapolation situation, as illustrated in Fig. 11.

In operational settings, non-exceedance frequencies of the lower (0.1 quantile) and upper (0.9 quantile) bounds of the predictive interval of the theoretical 80% coverage rate are of particular interest: it is expected that those values remain close to 10%. Figures 12 and 13 reveal that on average the 0.1 quantile is generally better assessed than the 0.9 quantile on average,







**Figure 8.** Distribution over the basins of the values of the log-sinh transformation parameters obtained during the calibration step for the four different lead times.

which is not the most desired behaviour for operational matters. More importantly, it can be seen that the lack of reliability of the log transformation seen for 3 LT in Fig.10 appears to be related to an underestimation of the exceedance frequency for the 0.1 quantile, while the non-exceedance frequency for the 0.9 quantile remains limited compared to the other transformations. These results highlight that reliability can have different complementary facets and that some parts of the predictive distributions can be more or less reliable. In a context of flood forecasting, particular attention should be given to the upper part of the predictive distribution.



### 3.2.2 Overall performance

In addition to reliability, we looked at other qualities of the probabilistic forecasts, namely the overall performance and accuracy. We also checked their sharpness. The distributions of four performance criteria are showed for lead time LT in Fig. 14. We note that the log transformation has the closest median value for the coverage ratio, at the expense of a lower median relative

5    sharpness value, because of larger predictive interval widths caused by the multiplicative behaviour of the log transformation. In addition, the CRPSS and the NSE distributions have limited sensitivity to the variable transformation, even if we can see that not using any transformation yields slightly better results. This confirms that the CRPSS itself is not sufficient to evaluate the adequacy of uncertainty estimation. Similar results were obtained for the other lead times (see the figures in Supplementary Material).







**Figure 9.** Examples of PIT diagrams obtained on the control data set D3, with different transformations at four locations: the Meu River at Montfort-sur-Meu (477 km$^2$): the forecasts are strongly biased; the Aa River at Wizernes (392 km$^2$): the uncertainty assessment is clearly under-dispersive; the Thérain River at Beauvais (755 km$^2$): the forecasts are reliable and the calibrated log-sinh and Box-Cox transformations (on D2$_{sup}$) are equivalent to the log transformation, which is here the best transformation on D3; the Sioulet River at Miremont (473 km$^2$): the calibration on D2$_{sup}$ leads to log-sinh and Box-Cox transformations equivalent to no transformation, which turns out not to be relevant on the control data set where the log and the Box-Cox transformations are more reliable.





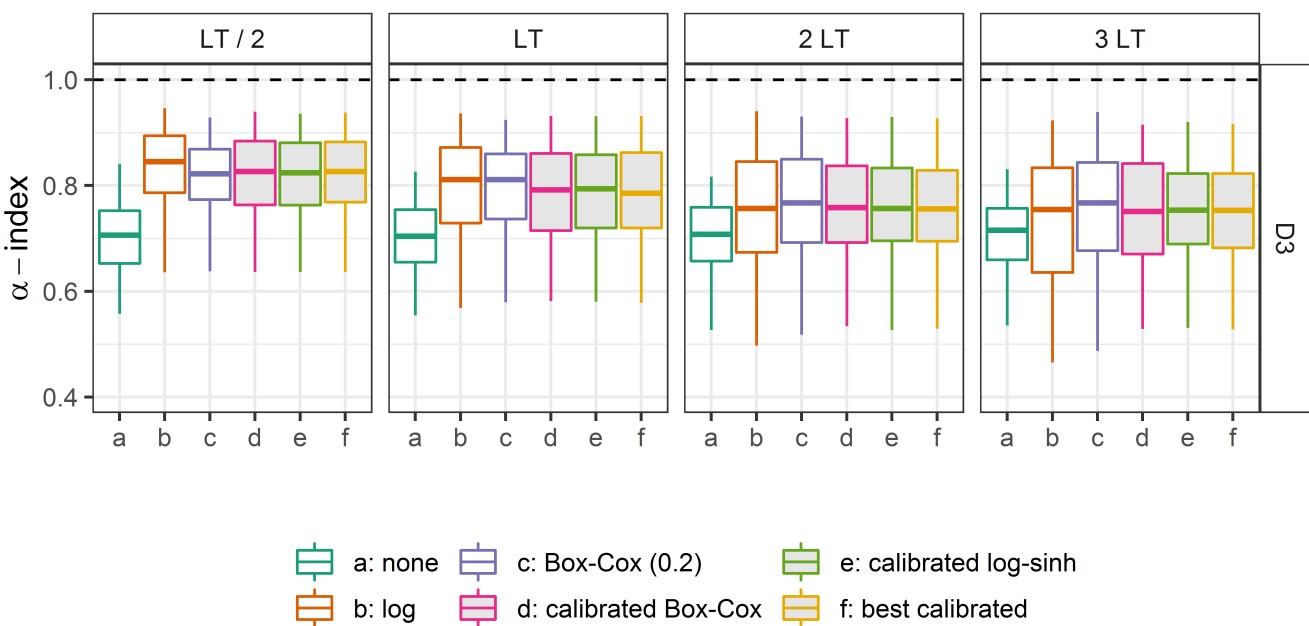

**Figure 10.** Distributions of the $\alpha$-index values on the control data set D3, obtained with different transformations for four lead times (the filled boxplots represent the calibrated distributions). Boxplots (5th, 25th, 50th, 75th and 95th percentiles) synthesise the variety of scores over the catchments of the data set.

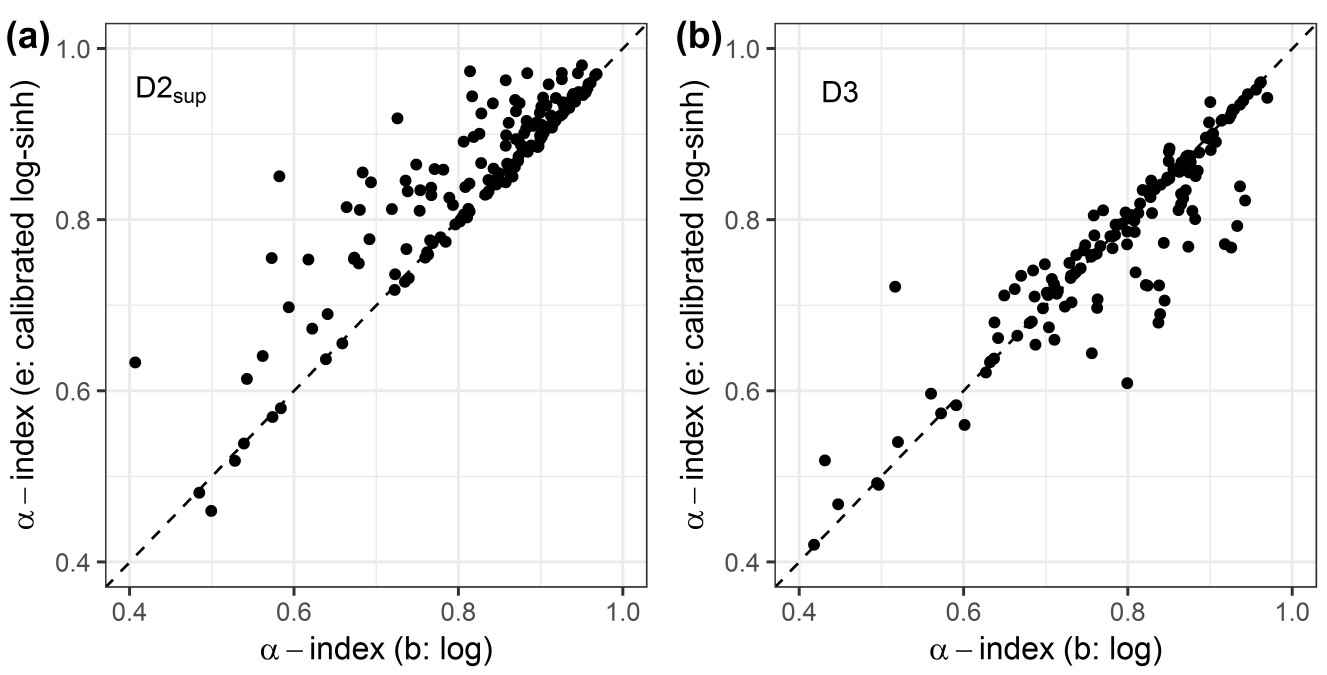

**Figure 11.** Scatter plots of the reliability $\alpha$-index obtained with the log transformation and the log-sinh transformation: (a) on D2$_{sup}$ in the calibration step and (b) on D3 in the control step.





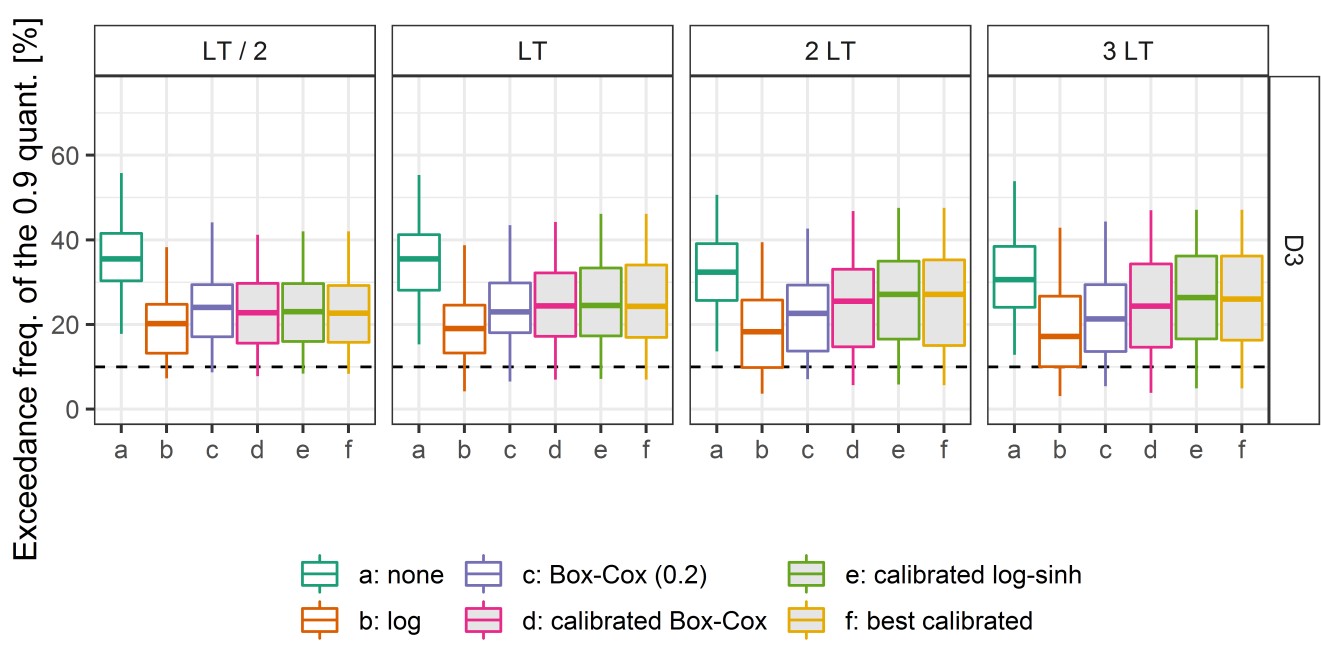

**Figure 12.** Distributions over the catchment set of the exceedance frequency of the 0.9 quantile on the control data set D3, obtained with the different transformations tested (the filled boxplots are related to calibrated transformations).





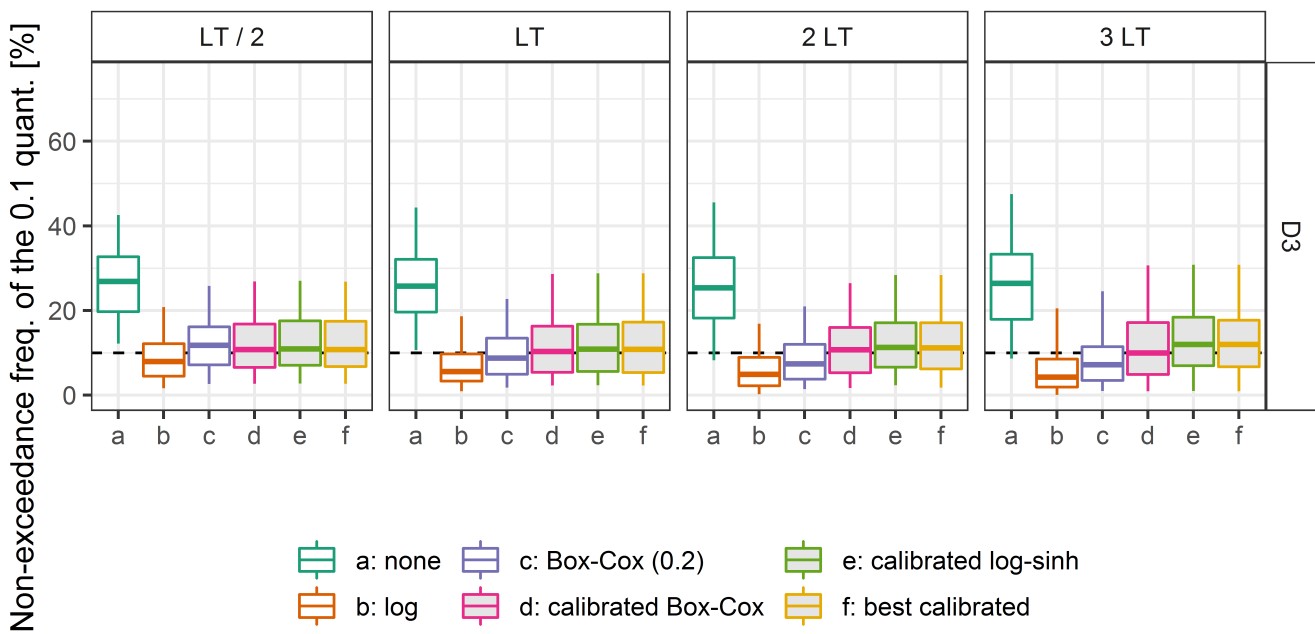

**Figure 13.** Distributions over the catchment set of the non-exceedance frequency of the 0.1 quantile on the control data set D3, obtained with the different transformations tested (the filled boxplots are related to calibrated transformations).





**Figure 14.** Distributions of coverage rate, relative sharpness, CRPSS and NSE values over the catchment set on the control data set D3, obtained with the different transformations tested (the filled boxplots are related to calibrated transformations).





## 4 Discussion

### 4.1 Effect of the number of parameters of the transformations

Overall, the results obtained on the control data set suggest that the log transformation and the fixed Box-Cox transformation ($BC_{\lambda=0.2}$) can yield relatively satisfactory $\alpha$-index and coverage ratio values given their multiplicative or near-

multiplicative behaviour in extrapolation. More tapered behaviours that can be obtained with the calibrated Box-Cox or log-sinh transformations do not show advantages when extrapolating high flows on an independent data set. In other words, what is learnt during the calibration of the more complex parametric transformations does not yield better results in an extrapolation context.

To investigate whether another calibration strategy could yield better results, we compared the performance on the D3 data

set, when the calibration is achieved on D2$_{\mathrm{sup}}$ ("f: best calibrated") or on D3 ("g: best reliability"). The results shown in Fig. 15 reveal that even when the best parameter set is chosen among the 217 possibilities tested in this study (17 for the Box-Cox and 200 for the log-sinh), the $\alpha$-index distributions are far from perfect and reliability decreases with increasing lead time. It suggests that the stability of the residuals distributions when extrapolating high flows might be a greater issue than the choice of the variable transformation. Nonetheless, the gap between the distributions of the non-parametric transformations ("b" and

"c"), the best calibrated transformation ("f") and the best performance that could be achieved ("g") highlights that it might be possible to obtain better results with a more advanced calibration strategy. This is, however, beyond the scope of this study and is therefore left for further investigations.

### 4.2 Performance loss in an extrapolation context

We investigated whether the reliability and reliability loss observed in an extrapolation context were correlated with any of

the properties of the forecasts. First, Fig. 16 shows the relationship between the $\alpha$-index values obtained in D2$_{\mathrm{sup}}$ and those obtained in D3 for three representative transformations. The results indicate that it is not possible to anticipate the $\alpha$-index values when extrapolating high flows in D3 based on the $\alpha$-index values obtained when extrapolating high flows in D2$_{\mathrm{sup}}$.

In addition, two indices were chosen to describe the degree of extrapolation: the ratio of the median of the forecasted discharges on D3 over the median of the forecasted discharge on D2$_{\mathrm{sup}}$ (Fig. 17), and the ratio of the median of the forecasted

discharges on D3 over the discharge for a return period of 20 years (rarity of the events of D3, for catchments where the assessment of the vicennial discharge was available in the national database: http://www.hydro.eaufrance.fr, not shown). In both cases, no trend appears, regardless of the variable transformation used, with Spearman correlation coefficients lower than 0.5. The reliability can remain high for some catchments even when the magnitude of the events of the control data set are much higher than those of the training data set.

Finally, Fig. 18 shows the reliability loss as a function of the relative accuracy of the deterministic forecasts on D2$_{\mathrm{sup}}$. A normalised RMSE was used to facilitate the visual representation, as in Lobligeois et al. (2014). Again, no clear trend is seen, which means that the goodness-of-fit during the calibration phase cannot be used as an indicator of the robustness of the uncertainty estimation in an extrapolation context.



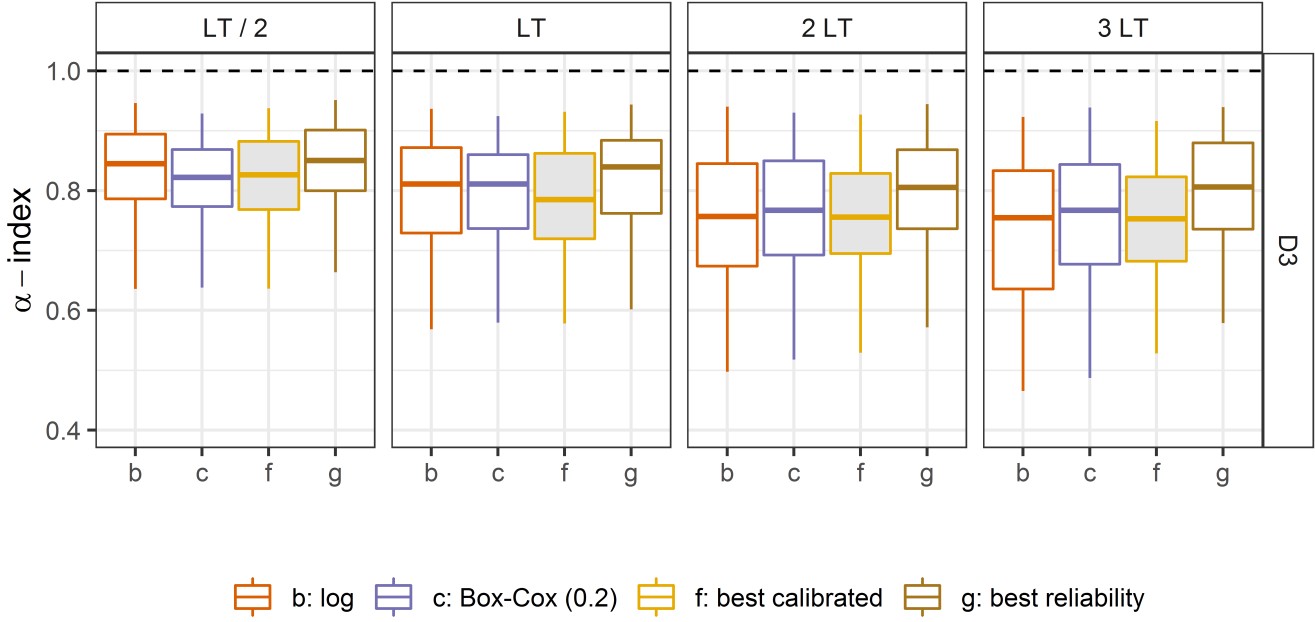

**Figure 15.** Distributions of the $\alpha$-index values on the control data set D3, obtained with different transformations for four lead times. Option "g" gives the best performance that could be achieved with this model and this post-processor for these catchments.

## 4.3 Empirical-based vs. distribution-based uncertainty assessment

Besides the reduction of heteroscedasticity, variable transformations may be used to fulfil the assumption of normality. Some post-processors are based on this hypothesis, such as the MCP or the meta-Gaussian model, and the NQT was designed to precisely achieve this. Here, we checked the normality of the residuals computed with the transformed variables using the

Shapiro-Francia test. For each parametric transformation, we selected the parameter set of the calibration grid which obtains the highest p-value. For more than 98% of the catchments, the p-value is lower than 0.018 (respectively, 0.023) when the Box-Cox transformation (respectively, the log-sinh transformation) is used. This indicates that there are only a few catchments for which the normality assumption is not to be rejected. In a nutshell, the variable transformations can stabilise the variance, but they do not necessarily ensure the normality of the residuals. It is important not to overlook this frequently encountered issue

in hydrological studies.

Even if there is no theoretical advantage to using the Gaussian distribution calibrated on the transformed-variable residuals rather than the empirical distribution to assess the predictive uncertainty, we tested the impact of this choice. For each transformation, the predictive uncertainty assessment obtained with the empirical transformed-variable residuals distribution is compared to the assessment based on the Gaussian distribution whose mean and variance are those of the empirical distri-

bution. Figure 19 shows the $\alpha$-index distributions obtained over the catchments for both options on the control data set D3.





**Figure 16.** Comparison of the $\alpha$-index values obtained in D2$_{sup}$ and D3. One point for each catchment. Similar results were obtained for the three other transformations (not shown).

We note that no clear conclusion can be drawn. No transformation (or identity transformation), which does not reduce the heteroscedasticity at all, benefits from the use of the Gaussian distributions for all lead times. In contrast, the predictive uncertainty assessment based on the empirical distribution with the log transformation is more reliable than the one based on the Gaussian distribution. For short lead times, it is slightly better to use the empirical distributions for the calibrated transformations (Box-Cox and log-sinh), but we observe a different behaviour for longer lead times. For these longer lead times, assessing the predictive uncertainty by the Gaussian distribution fitted on the empirical distributions of transformed residuals obtained with the calibrated log-sinh or Box-Cox transformations is the most reliable option. It is better than using the log transformation with the empirical distribution, but not very different from using the $BC_{\lambda=0.2}$ transformation.





**Figure 17.** Reliability loss as a function of the rarity of the D3 events (ratio of the mean forecasted discharge D3 to mean forecasted discharge D2$_{\text{sup}}$ over the catchment set. Similar results were obtained for the three other transformations (not shown).

To further investigate the impact of the choice between the empirical or the Gaussian distributions, Figure 20 shows the scatter plots of the $\alpha$-index values obtained with the empirical distribution and with the Gaussian distribution for a lead time equal to the lag time. The values obtained for all catchments are well distributed on both sides of the bisector for the two transformations shown here and for the others (see the figures in Supplementary Material). There is no systematic behaviour:

5 for some transformations, it is slightly better to choose the theoretical Gaussian quantiles, while empirical quantiles provide slightly more reliable predictive uncertainty assessment for others. It is important to note that the choice of the distribution is not the dominant factor: the variability of the distance to the bisector is much lower than the $\alpha$-index variability obtained among catchments. The same pattern is observed for the other lead times that are not shown here (see the figures in Supplementary Material).





**Figure 18.** Reliability loss as a function of the relative accuracy of the deterministic forecasts on D2$_{sup}$. Similar results were obtained for the three other transformations (not shown).

## 4.4 Links to previous results

As McInerney et al. (2017) pointed out, the choice of the heteroscedasticity modelling has a great impact on the predictive performance of probabilistic forecasts. Our results show that in an extrapolation context the log transformation provides overall the best results on the control data set for most catchments. The calibrated transformations are close to the log transformation for a significant number of catchments. Interestingly, for some catchments, the calibrated transformations are close to the no transformation on D2$_{sup}$, whereas the best choice on the control data set would have been another transformation. The $BC_{\lambda=0.2}$ transformation also yields reasonable results for a significant number of catchments. Nevertheless, for some sites, the best transformation is close to no transformation.







**Figure 19.** Distributions of the $\alpha$-index values on the control data set D3, obtained with different transformations for four lead times, when using the empirical residuals distributions (straight boxplots) and the Gaussian distributions (dashed boxplots).





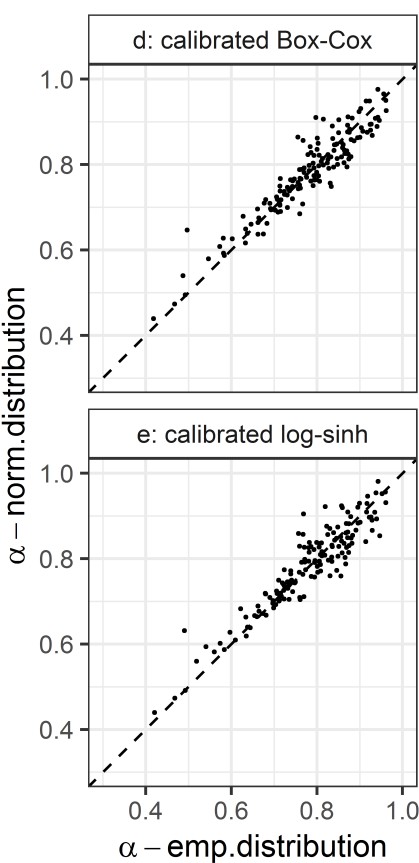

**Figure 20.** Scatter plots of the $\alpha$-index values obtained over the catchment set for two transformations, for the lead time LT: x-axis, using the empirical residuals distribution observed on the training data set; y-axis: using the Gaussian distribution fitted on the previous one.





## 5    A need for a focus change?

In most modelling studies, several methodological steps depend on the range of the observations. First, calibration is designed to limit the residual errors in the available historical data. However the largest residuals are often associated with the highest discharge values. It is well known that removing the largest flood events from a data set can significantly modify the resulting

calibrated parameter set. This is particularly true with the use of some common criteria such as quadratic criteria which strongly emphasise the largest errors (Legates and McCabe Jr., 1999; Berthet et al., 2010; Wright et al., 2015). Conversely, it is likely that "unavailable data" such as a physically realistic but (so far) unseen flood would significantly change the calibration results if it could be included in the calibration data set. Moreover, model conceptualisation (building) itself is often based on the understanding of how a catchment behaves "on average". In some studies, outliers may even be considered as disturbing and

be discarded (Liano, 1996).

However, to provide robust models for operational purposes, we also need to focus on rare (rarely observed) events, still keeping in mind all the well-known issues associated with working with (too) few data (Anctil et al., 2004; Perrin et al., 2007). For predictive uncertainty assessment, this issue is exacerbated by the seasonality of hydrological extremes (Allamano et al., 2011; Li et al., 2013) for most approaches, which rely heavily on data (beyond data-learning approaches, all models

which need to be calibrated). Therefore, there is an urgent need to gather and compile data on extreme events (Gaume et al., 2009). Nevertheless, operational forecasters must still prepare themselves to work in an extrapolation context, as pointed out by Andréassian et al. (2012).





## 6 Conclusions

Even if major floods are rare, it is of the utmost importance that the forecasts issued during such events are reliable to facilitate an efficient crisis management. Like Lieutenant Drogo in the Tartar Steppe who spent his entire life fulfilling his day-to-day duties, but waiting in his fortress for the invasion by foes (Buzzati, 1940), many forecasters expect and are preparing for a major event, even if their routine involves only minor events. That is why a strong concern for the extrapolation context should be encouraged in all modelling and calibration steps. This article proposes a control framework focusing on the forecasting performance in an extrapolation context.

We use this framework to test the predictive uncertainty assessment using a statistical post-processing of a rainfall-runoff model, based on a variable transformation. The latter has to handle the heteroscedasticity of the predictive uncertainty, which is very problematic in an extrapolation context to issue reliable uncertainty assessment. As pointed out by McInerney et al. (2017), the choice of the heteroscedastic error modelling approach makes a significant impact on the predictive performance. This is true as well in an extrapolation context. The main finding is that the more parametric transformations do not achieve significantly better results than the non-parametric transformations:

- while it allows a flexibility which can theoretically be very attractive in an extrapolation context, the log-sinh transformation is not more reliable in such a context;

- the non-parametric log transformation and Box-Cox transformation with the $\lambda$ parameter set at 0.2 are robust and compare favourably.

The findings reported herein corroborate the results of McInerney et al. (2017) within the context of flood forecasting and extrapolation: calibrating the Box-Cox or log-sinh transformation can be counter-productive. We therefore suggest that operational flood forecasting services could consider the less flexible but more robust options: using the log transformation or the Box-Cox transformation with its $\lambda$ parameter set at 0.2 or between 0.1 and 0.3.

Importantly, these results reveal significant performance losses on some catchments when it comes to extrapolation, whatever variable transformation is used. Even if the scheme tested yields satisfying results in terms of reliability for the majority of catchments, it fails on a significant number of catchments and further investigations are needed to gain a deeper understanding of when and why failures occur. There are also some perspectives, for example with the regionalisation of the predictive distribution assessment, as proposed in Bourgin et al. (2015) and Bock et al. (2018), which could help build more robust assessment of uncertainty quantification when forecasting high flows.





## Appendix A:  GRP model

The GRP model belongs to the suite of GR models (Michel, 1983). These models are designed to be as simple as possible but efficient for hydrological studies and for various operational needs, resulting in parsimonious model structures (https://webgr. irstea.fr/en/models/a-brief-history/, last access: 1 April 2019). The GRP model is designed for forecasting purposes (Berthet,
2010). It is a deterministic continuous lumped storage-type model (Fig. A1). The inputs are limited to areal rainfall and (interannual) potential evapotranspiration (both data may be available in real time). It can be understood as the sequence of two hydrological functions:

- a production function which is the same as in the well-known GR4J model developed by Perrin et al. (2003);

- a routing function which is a simplified version of the GR4J's routing function, since it only counts one flow branch
composed of a unit hydrograph and quadratic routing store. The tests showed that the performance of the GRP and GR4J structures was similar in a forecasting mode.

A snow module (Valéry et al., 2014) may be implemented on top of the model if necessary.

Like any GR model, it is parsimonious. It has only three parameters: (a) an adjustment factor of effective rainfall which contributes to finding a good water balance, (b) the unit hydrogram time base used to account for the time lag between rainfall
and streamflow and (c) the capacity of the routing store, which temporally smooths the output signal.

Its main difference with the other GR models is the implementation "in the loop" of two data assimilation schemes:

- a state updating procedure which modifies the main state of the routing function as a function of the last discharge values;

- an output updating based on an autoregressive model of the multiplicative error or an artificial neural network (multi-layer perception) whose inputs are the last discharge value and the two last forecasting errors. In this study, the autoregressive
model was used.

The parameters are calibrated in forecasting mode, i.e., with the application of the updating procedures. This model is used by the French Flood Forecasting Services, some hydroelectricity suppliers and canal managers at the hourly time step in order to issue real-time forecasts for lead times ranging from a few hours to a few days at several hundred sites. Recently, Ficchì et al. (2016) pave the way to a multi-time step GRP version.

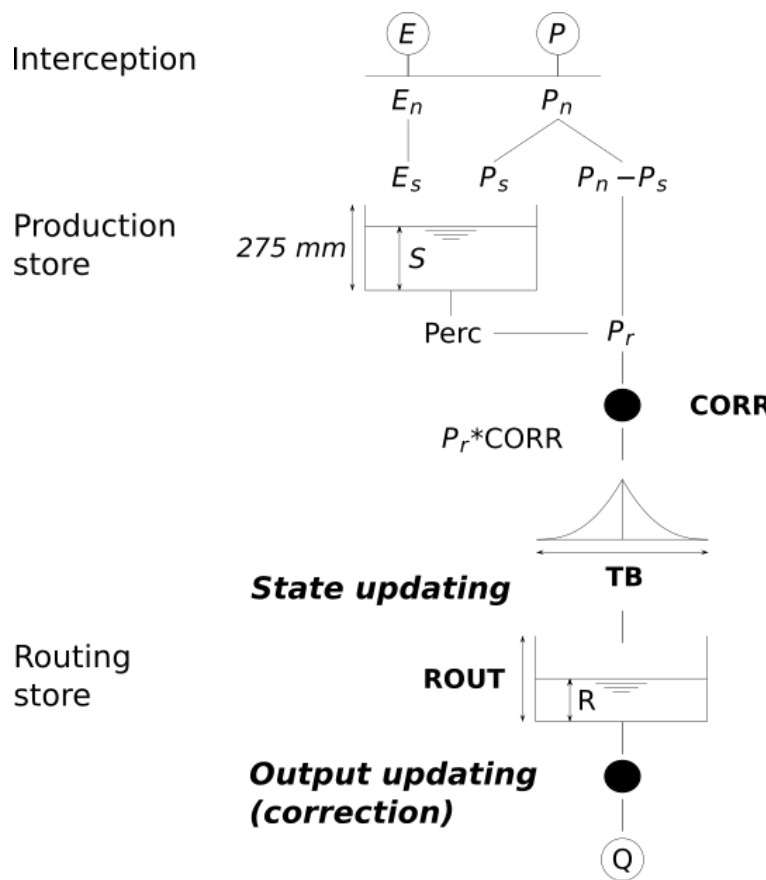

**Figure A1.** The GRP model flow-chart. After an interception step, the production function splits the net rainfall, according to the level of the production store. The effective rainfall is the sum of the direct flow and the percolation from this store. A corrective multiplicative coefficient is then applied. Then the flow runs through a unit hydrograph and the routing store.





**Table B1.** Summary of the different behaviours of the log-sinh transformation.

| Cases | Behaviours |
|---|---|
| $\alpha > 3 \cdot \beta$ | Additive error model |
| $y > 3 \cdot \beta$ | Additive error model |
| $\alpha \ll y \ll \beta$ | Multiplicative error model |
| $y \ll \alpha \ll \beta$ | Additive error model |
| Otherwise if $\alpha + y \ll \beta$ | Multiplicative error model (with an additive constant) |

## Appendix B: The log-sinh transformation behaviours

### B1  Log-sinh transformation formulations

In this study, we used the formulations of the log-sinh transformation chosen by Del Giudice et al. (2013):

$$g_{\alpha,\beta} : y \mapsto \beta \cdot \log\left(\sinh\left(\frac{\alpha + y}{\beta}\right)\right) \tag{B1}$$

5   It is strictly equivalent to the formulation used by McInerney et al. (2017):

$$g'_{a,b} : y \mapsto \frac{1}{b} \log\left(\sinh\left(a + b \cdot y\right)\right) \tag{B2}$$

### B2  Different behaviours

Depending on the relative values of $\alpha$ and $\beta$ and on the value range of $y$, compared to $\alpha$ and $\beta$, the log-sinh transformation can be reduced to an affine transformation (i.e. $g_{\alpha,\beta}\left(y + \varepsilon\right) - g_{\alpha,\beta}\left(y\right) = \delta \cdot \varepsilon$) or to the log transformation. The former case is

10   equivalent to no transformation (identity function; additive error model), whereas the latter one is equivalent to a multiplicative error model (Table B1).

When $y \ll \beta$,

$$
\begin{aligned}
\sinh\left(\frac{\alpha + y}{\beta}\right) &= \sinh\left(\frac{\alpha}{\beta}\right) \cdot \cosh\left(\frac{y}{\beta}\right) + \\
&\quad \sinh\left(\frac{y}{\beta}\right) \cdot \cosh\left(\frac{\alpha}{\beta}\right) \\
&\simeq \sinh\left(\frac{\alpha}{\beta}\right) \cdot \left(1 + \frac{y^2}{2 \cdot \beta^2} + \ldots\right) + \\
&\quad \cosh\left(\frac{\alpha}{\beta}\right) \cdot \left(\frac{y}{\beta} + \ldots\right)
\end{aligned}
$$

Thus,

$$g_{\alpha,\beta}\left(y\right) \simeq \beta \cdot \log\left(\sinh\left(\frac{\alpha}{\beta}\right) + \cosh\left(\frac{\alpha}{\beta}\right) \cdot \frac{y}{\beta}\right)$$





When $\alpha \ll \beta$, the latter results in:

$$g_{\alpha,\beta}(y) \simeq \beta \cdot \log\left(\frac{\alpha + y}{\beta}\right) \tag{B3}$$

### B2.1 Cases where the log-sinh transformation is equivalent to an affine transformation

If $x > 3$, $\sinh(x) \approx e^x/2$, then $\log(\sinh(x)) = x - \log(2)$. Therefore when $z = (\alpha + y)/\beta > 3$, the log-sinh transformation is equivalent to an affine transformation. In such cases, $g_{\alpha,\beta}^{-1}(g_{\alpha,\beta}(y) + \varepsilon) = y + \varepsilon$.

This happens when

– $\alpha > 3 \cdot \beta$;

– the $\beta$ value is chosen large enough so that for any $y$ value in the discharge range, $y > 3 \cdot \beta$.

Moreover, when $y \ll \alpha \ll \beta$, (B3) gives:

$$g_{\alpha,\beta}(y) \simeq \beta \cdot \left[\log\left(\frac{\alpha + y}{\alpha}\right) + \log(\alpha) - \log(\beta)\right]$$
$$\simeq \frac{\beta \cdot y}{\alpha} + \beta \cdot (\log(\alpha) - \log(\beta))$$

The log-sinh transformation is then equivalent to an affine transformation.

### B2.2 Cases where the log-sinh transformation is equivalent to a log transformation

When $\alpha \ll y \ll \beta$, (B3) gives:

$$g_{\alpha,\beta}(y) \simeq \beta \cdot \log\left(\frac{y}{\beta}\right)$$

The log-sinh transformation is then equivalent to a mere log transformation.

## B3 Calibration

The $\alpha$ and $\beta$ parameters are in the same physical dimension as the $y$ variable. Since this study is dedicated to the extrapolation context, we used the following adimensional parametrization to calibrate the variable transformation on various catchments. $\alpha$ and $\beta$ are compared to the maximum forecasted discharge on the $D2_{\text{sup}}$ data subset:

$$\alpha = \gamma_1 \cdot \max_{D2}\left(\widetilde{Q}\right) \text{ and } \beta = \gamma_2 \cdot \max_{D2}\left(\widetilde{Q}\right) \tag{B4}$$

In the calibration step, the parameter space is explored on a $(\gamma_1, \gamma_2)$ grid: 18 values of $\gamma_1$ from 0.01 to 100 and 15 values of $\gamma_2$ from 0.1 to 100 were tested, excluding combined values leading to the very same behaviours, such as $\gamma_1 \gg 3 \cdot \gamma_2$, equivalent to no transformation (additive error model). A total of 200 $(\gamma_1, \gamma_2)$ combinations were tested for each calibration.



**Appendix C: Supplementary material**





**Figure C1.** Distributions of coverage rate, relative sharpness, CRPSS and NSE values over the catchment set on control data set D3, obtained with the different transformations tested (the filled boxplots are related to calibrated transformations), for lead time LT / 2.





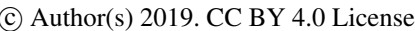

**Figure C2.** Distributions of coverage rate, relative sharpness, CRPSS and NSE values over the catchment set on control data set D3, obtained with the different transformations tested (the filled boxplots are related to calibrated transformations), for lead time 2 LT.





**Figure C3.** Distributions of coverage rate, relative sharpness, CRPSS and NSE values over the catchment set on control data set D3, obtained with the different transformations tested (the filled boxplots are related to calibrated transformations), for lead time 3 LT.





**Figure C4.** Scatter plots of the reliability $\alpha$-index over the catchment set: x-axis, using the empirical residuals distribution observed on the training data set; y-axis: using the Gaussian distribution fitted on the previous one.



*Author contributions.* All co-authors contributed to and edited the paper.

*Competing interests.* The authors declare that they have no conflict of interest.

*Acknowledgements.* The authors thank Météo-France for providing the meteorological data and Banque HYDRO data base for the hydrological data. The contribution of the authors from Irstea was financially supported by SCHAPI (Ministry of the Environment)




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
