# Peer review of "A crash-testing framework for predictive uncertainty assessment when forecasting high flows in an extrapolation context"

_Hydrology and Earth System Sciences, 2019_

## Referee Comment (RC1) · Anonymous Referee #1 · 17 Jun 2019

**Review for "A crash-testing framework for predictive uncertainty assessment when forecasting high flows in an extrapolation context"**

**General comments**

This paper presents an approach for calibrating and evaluating extrapolated probabilistic hydrological predictions in the context of flood prediction. The authors consider a range of transformations for use in an uncertainty processor, and perform analysis over a large number of catchments, using multiple metrics to evaluate performance of the forecasts. The authors find that more complex transformations, which require calibration of parameters, may perform better over a calibration data-set, but typically do not perform best in an extrapolation context.

This is an interesting paper on an important topic, and is particularly relevant with a changing climate, where larger flooding events outside the range of historical observations may occur. The evaluation is comprehensive (large number of catchments, multiple metrics) and their analysis supports the key findings. However, I found that (i) the description of the uncertainty processor, and in particular the role of the transformations, was insufficient, and (ii) the discussion section requires additional work to explain the motivation for the additional analysis in this section. Therefore, I recommend major revisions be made to this article before it can be published in HESS.

**Specific comments**

**More details of EHUP**. The empirical hydrological uncertainty processor (EHUP) and the different transformations are described in Section 2.1.3 and 2.1.4, respectively. Unfortunately, the level of detail provided by the authors was not sufficient to allow me to understand how the EHUP works and how the transformations fit inside the EHUP. In particular,

- It is unclear how the transformations fit in with EHUP. A diagram or mathematical equations would help make this clearer.
- It is not clear what are the inputs and outputs of the EHUP.
- Why does the EHUP require a separate training period to transformation calibration period?

**Discussion section.** The motivation for a lot of the analysis performed in this section is not clear to me, doesn't seem to fit in with the aims of the study, and often does not seem to match the sub-headings

- **Section 4.1**
  - **Pg 25, lines 9-17:** This paragraph doesn't seem to be addressing the heading of the section, which is on the number of parameters in the transformation.
- **Section 4.2**
  - This sub-section seems like it is attempting to determine what the key drivers for performance are, but this is not evident from heading.
  - Since the authors did not find any key drivers for poor performance, I'm not sure if this analysis adds much value.
- **Section 4.3**
  - The motivation for this section is unclear to me. Why are you comparing empirical and distribution based uncertainty? This seems tangential to the aims of the study. A brief sentence at the start to explain what you're looking into, and why, would be useful.

- You are comparing "empirical-based" and "distribution-based" uncertainty assessment in this section. Since you have not explained the EHUP in enough detail, it is not clear which of these 2 approaches you have used for the rest of this study.
- **Section 4.4**
  - This section is about making links to previous studies, but you cite only one paper and make no comparison to the findings of that paper.
- **Section 5:** "A need for a focus change."
  - This section is not long enough for a separate section, and is a discussion topic. I suggest moving this to the discussion.
- **Limitations and future work**
  - It would be useful to have a sub-section discussing the limitations of this study and future research.

**Too many figures**. I believe this paper has too many figures. I recommend
- Merging some figures
  - fig 6 and 10
  - fig 12 and 13
- Is there any point in showing all 3 transformations for fig 16-18?
  - You could consider a single transformation and combine into a single figure.
  - Or you could move fig 17-18 to sup mat since they don't show any correlations.

**Figure 5:** I like the idea of having a diagram to explain how the different sets D1, D2sup, D2inf and D3 are used in calibration and evaluation, but I found this figure particularly confusing. In particular,
- Why is different data used for EHUP in calibration and evaluation?
- Why does D1 have many more points than D2sup and D3? From the text I thought D2sup and D3 had 720 points, while D1 had 500 points?
- What's the purpose of showing the residuals on the y-axis? These are not discussed in the text.
- In panel b, most of the points for D2inf (light blue) are hidden behind points for D1 (red).

**Technical corrections**

**Abstract:** "… the Box-Cox transformation with a parameter between 0.1 and 0.3 can be a reasonable choice for flood forecasting"

You have only shown results for lambda=0.2 in this paper. How you can say that using lambda between 0.1 and 0.3 can be a reasonable choice?

**Table 1:** Change "percentiles" to "quantiles"

**Pg 5, line 10:** "For each catchment, the lag time LT is estimated as the lag time maximising the cross-correlation between rainfall and discharge time series."

What is the relevance of estimating LT? This becomes more obvious later in the paper, but should be described briefly here.

**Pg 5, line 15:** "It is a deterministic lumped storage-type model that uses catchment areal rainfall and PE as inputs"

What rainfall is used to produce the GRP forecasts? Is observed rainfall used, forecast rainfall, etc? If it is observed rainfall, then how is this used in a forecasting context?

**Pg 6, line 3-4:** "Since herein only the ability of the post-processor to extrapolate uncertainty quantification is studied, the model is calibrated in forecasting mode over the 10-year series by minimising the sum of squared errors for a lead time taken as the lag time LT."

What is meant by "forecasting mode" here? More details on how forecasts are generated would be useful.

**Pg 7, lines 8-12:** Is the NQT actually used in this study? If so, it's not clear how and where it's used.

**Pg 7, lines 14-15:** If the NQT requires additional assumptions for the tails, how do you handle this problem in in this study?

**Pg 7, lines 30-31:** "McInerney et al. (2017) obtained their best results with lambda = 0.2 over 17 perennial catchments."

What do you mean by "best results"? Please provide some context for this statement.

**Pg 8, line 3:** Why does this equation use different notation than other transformations?

**Pg 10, line 6:** "maximum discharge of time series"

Make it clear you are referring to *forecast* discharge here.

**Pg 10, line 8:** "the first time step"

What do you mean by "first time step"? Do you mean the closest time step?

**Pg 10, lines 20-21, Pg 11, line 1**: The purpose of the "control", "training", and "calibration" subsets has not been explained. Please describe what they are used for.

**Pg 12, line 1:** It is unclear what the "coverage rate of the 80% predictive intervals" is. Please provide equations or description.

**Pg 13, line 6:** "i.e. from the distribution of the observed discharges over the events selected"

What is meant by "events selected"? Is this all events in G1, G2 and G3?

**Pg 15, lines 6-8:** "as expected, and that there is no significant difference between the calibrated Box-Cox transformation (d), the calibrated log-sinh transformation (e) and the best performing transformation (f)."

How are you determining whether differences between results are "significant"? A statistical test should be used to determine whether differences are "significant".

**Pg 15, line 8:** Is "best performing" the same as "best calibrated"? If so, use a single term.

**Pg 15, lines 9-10:** "Interestingly, the log transformation provides the best results for the other criteria (not used as the objective function)."

Are these results shown anywhere? If so, provide reference to figure.

**Pg 15, line 13-14:** "While the log-transformation behaviour is frequently chosen for LT/2 and LT, the additive behaviour becomes more frequent for 2 LT and 3 LT."

It is unclear what you mean by "additive behaviour" and how this is seen in the figures (i.e. what parameters relate to additive behaviour).

**Pg 17, lines 7-8:** "This confirms that the CRPSS itself is not sufficient to evaluate the adequacy of uncertainty estimation"

Similar findings about CRPS being insensitive to chosen data transformation have been made in other studies, e.g. Woldemeskel et al. (2018). It might be worth mentioning this.

**Figure 9 caption:** "Thérain River at Beauvais (755 km2): the forecasts are reliable and …."

This statement does not seem correct. I would say the forecasts are not reliable for "none".

**Figure 14:** Legend is missing

**Pg 25, line 22-23:** "The results indicate that it is not possible to anticipate the alpha-index values when extrapolating high flows in D3 based on the alpha-index values obtained when extrapolating high flows in D2sup."

There appears to be some correlation in Figure 16. What is the Spearmen correlation?

**Pg 25, lines 27-28:** "In both cases, no trend appears, regardless of the variable transformation used, with Spearman correlation coefficients lower than 0.5."

A Spearman correlation of 0.5 does not seem correct. If it was 0.5, then there would be a clear trend.

**Pg 25, line 31:** What is a "normalized RMSE" and why is it used? A sentence/equation describing this would be useful (rather than just a citation).

**Pg 26, line 11:** "Even if there is no theoretical advantage to using the Gaussian distribution calibrated on the transformed-variable residuals rather than the empirical distribution to assess the predictive uncertainty, we tested the impact of this choice."

If there is no theoretical advantage, why are you testing this?

**Pg 33, line 21:** "the Box-Cox transformation with its lambda parameter set at 0.2 or between 0.1 and 0.3."

You have only shown results for lambda=0.2 in this paper. How you can recommend using other values of lambda between 0.1 and 0.3 in the conclusions of this paper?

**Section B2.2.** This relationship was shown by McInerney et al. (2017) (Appendix A)

**References**

McInerney, D., Thyer, M., Kavetski, D., Lerat, J. & Kuczera, G. 2017. Improving probabilistic prediction of daily streamflow by identifying Pareto optimal approaches for modeling heteroscedastic residual errors. *Water Resources Research,* 53.

Woldemeskel, F., McInerney, D., Lerat, J., Thyer, M., Kavetski, D., Shin, D., Tuteja, N. & Kuczera, G. 2018. Evaluating residual error approaches for post-processing monthly and seasonal streamflow forecasts. *Hydrol. Earth Syst. Sci. Discuss.,* 2018**,** 1-40.

---

## Referee Comment (RC2) · Kolbjorn Engeland (Referee) · 21 Jul 2019

The paper presents a framework aiming at evaluating the performance of probabilistic forecasts on highest flood events that the post-processors are not calibrated for. The authors combine an empirical hydrological post-processor (EHUP) with different transformations, and compare the performance of the predictive distributions for forecasted floods that are higher than the floods used for training/calibrating the EHUP and the transformations.

The paper is interesting and deserves publication following a major revision. Below I list some important issues to be addressed in the revised manuscript.
Throughout the introduction, the importance of modelling the heteroscedasticity of the predictive uncertainty distribution emphasized. I miss a good argument why it is important (i.e. to obtain reliability), and you could refer to literature that shows this (i.e. McInerney et al., 2017). In the introduction and discussion, you ignore that other properties of the predictive distribution (i.e. bias and skewness) could also depend on forecasted flows. My experience is that a calibrated hydrological model tends to underestimate flood peaks, introducing a possible bias. Bremnes et al (2019) shows that the skewness depends on the predicted wind and that adding this property improves the forecasts for high wind speeds. You discuss this briefly in lines 4-9 on page 8. Is it possible that the results presented in Figures 12 and 13 indicate that the skewness is an important issue for the reliability of the predictive distributions, and that your approach has to small skewness?

I miss an explanation of which meteorological products you used to generate the discharge forecasts. Did you use the reanalysis mentioned in 2.1.1 or did you use a forecast product? The EHUP needs a better description, in particular how it is used in combination with the different transformations. I also need a clarification of which data were used for estimating the empirical quantiles of errors. On page 6 you write that the top 5% pairs ranked by simulated values are used, whereas on page 11 you write that the subsets D1 and D2 were used. Figure 5 indicates that not the whole D1 subset was used for training of the EHUP, only the highest discharge values. A consistent description is needed to avoid confusion.

The discussion section needs a better organization. Results presented in section 4.1 could be integrated into section 3. In Figure 15, the only new result is the boxes labelled 'g'. Could it be integrated into Figure 10? Section 4.2 and 4.3 introduces new results that do not directly relate to the objectives / questions listed on Page 4. If these results should be included, you could add one more objective related to these results, and integrate the results into Section 3. I suggest to exclude results and discussion in section 4.3 (including Figure 19 and 20) and only briefly summarize the main findings.
Section 5 could be also a part of the discussion.

The number of figures could be reduced. Figure 2 – right panel is not necessary. Figure 4a and 4b could be merged. Is it possible the merge Figure 5a and b? Could results in Figure 15 be included in Figure 10? Figure 11 is not necessary. Figure 12, and 13 could be merged. I suggest to remove Figure 14a since it is just another measure of reliability and does not add new information to the results. Figures 19 and 20 could be excluded or moved to supplementary materials.

Below follows some detailed comments to the manuscript:

Table 2: When you compare discharge across catchments, I think it is better to use specific discharge (I/s/km2).

Figure 3: What is the explanations for this apparently negative skewness for the predictive distribution? The log-transformation leads to slightly positively skewed predictive distribution?

Figure 14: Legend is missing

Page 2: The meaning of the first paragraph of section 1.2 is difficult to understand. In particular the two first sentences needs more context.

Page 3: I suggest to write the first paragraph of 1.2.1 as:

"A first approach that intends to model each source of uncertainty separately and to propagate these uncertainties through the modelling chain is presented in Renard et al., (2010). According to this approach, the heteroscedasticity of the predictive uncertainty distribution results from the separate modelling of each source of uncertainty and from the statistical model specification. While this approach is promising, operational application can be hindered by the challenge of making the hydrological modelling uncertainty explicit, as pointed out by Salamon and Feyen (2009)."

Question or the paragraph above: which statistical model needs to be specified? Is it

HESSD
for the meteorological input or for the simulated discharge?

Page 4 lines 7-8: These approaches are not exclusive of each other. Even when future precipitation is the main source of uncertainty, postprocessing is often required to produce reliable hydrological ensembles Question: What does 'these approaches' refer to? does it refer to all approaches presented in the introduction or all approaches presented in section 1.2.2?

Page 5 Section 2.1.1 : Maye a question of style, you write 'We used a set of 154 unregulated catchments spread throughout France (Fig. 1) to test our hypotheses over various hydrological regimes and forecasting contexts." Since you have chosen to use formulate research questions and not to test hypotheses in this paper, the sentence could be changed to 'We used a set of 154 unregulated catchments spread throughout France (Fig. 1) over various hydrological regimes and forecasting contexts to provide robust answers to our research questions.

Page 7, line 19: You write that the log-transformation is non-parametric. I would rather say it is a parametric transformation with no tuning parameters. The term non-parametric is often used when you make no assumptions about the form or parameters of the transformation.

Page 10 Section 2.2.2. How did you select more than one event? According to the description you selected one event defined by the maximum discharge of the time series.

Page 11: Why has the calibration data subset to encompass time steps with simulated discharge values higher than those of the training subset?

Page 13: First equation: define k and N Second equation: Could you use the same notation as in the first equation. i.e. write it as sum divided by number of time steps?

Page 15, lines 9-10: Here you comment results that are not yet presented, making it difficult for the reader to follow. I think this sentence fits better in the discussion

**HESSD**
Page 16: The last three lines have to be re-phrased in order to make sense: "In operational settings, non-exceedance frequencies of the lower (0.1 quantile) and the exceedance frequencies of the upper (0.9 quantile) bounds of the predictive distribution are of particular interest. It is expected that those values remain close to 10% for a reliable predictive distribution. Deviations from these frequencies indicates biases in the estimated quantiles."

Page 17 lines 3-5: I think it is better to write something like this (I think it is better to write that the 0.1 and 0.9 quantiles are over or under-estimated, and not the (non)-exceedance frequency of the (0.1) and 0.9 quantiles.): "More importantly, it can be seen that the lack of reliability of the log transformation seen for 3 LT in Fig.10 appears to be related to an underestimation of both the 0.1 and 0.9 quantile. Compared to the other transformations, the log transformation has the largest under-estimation of the 0.1 quantile and the smallest under-estimation of the 0.9 quantile."

Page 18 Section 3.2.2: Please be more precise in the comments: What is 'overall performance' ? Suggestion for re-phrasing some of the sentences:

"We note that the log transformation has the highest median value for the coverage ratio, and is also the closest to the 80% ratio that is expected from a reliable forecast,"

"In addition, the CRPSS and the NSE distributions have limited sensitivity to the variable transformation. We can even see that not using any transformation yields slightly better results according to NSE."

Page 33: Please provide clear conclusions related to each of the objectives and answer the research questions asked in the introduction.

New reference in this review: Bremnes, J.B., 2019: Constrained Quantile Regression Splines for Ensemble Postprocessing. Mon. Wea. Rev., 147, 1769–1780, https://doi.org/10.1175/MWR-D-18-0420.1

181, 2019.

---

## Author Comment (AC2) · 10 Aug 2019

The reply to both referees is in the attached file (same file as for the 1st referee.

Respectfully yours,

Please also note the supplement to this comment:
https://www.hydrol-earth-syst-sci-discuss.net/hess-2019-181/hess-2019-181-AC2-supplement.pdf
* * *
181, 2019.

---

## Editor Comment (EC1) · Dimitri Solomatine (Editor) · 17 Aug 2019

The authors cover an important topic, presenting an approach to calibrate and evaluate the probabilistic flood predictions, when considered outside of the historical data range ("crash test"). This is indeed very important in the context of global/climate changes, and with current attention of researchers to explicitly account for uncertainty and to model verification under changing conditions. Experimental work and analysis is comprehensive and valuable. I agree with referess that oragnisation of the paper, and at places, clarity could be improved, and the EHUP processor needs a better presentation. They also advise the very relevant references, to ensure better link to earlier work,

and for a wider context. The authors' replies show that the authors have a clear plan for revision, and I wish them good luck.

———————————————————

---

## Author Response (AR1)

**Reply to the review comments on the manuscript "A crash-testing framework for predictive uncertainty assessment when forecasting high flows in an extrapolation context" by Berthet *et al.* (manuscript hess-2019-181)**

We first thank both referees for their detailed reviews and analyses of the submitted article. We also thank them for their positive opinion about the scientific soundness of this study and their constructive comments. They are very valuable to improve the manuscript and we intend to follow most of them (see details hereafter).

Both referees share some comments:

1. In the description of the methodology, the empirical hydrological uncertainty processor (EHUP) needs a better and more detailed description. Indeed, since there are some (quoted) references, we drastically reduced this description. It is clearly not sufficient and we propose to provide a more detailed description in a revised version.

2. The discussion deserves a better organization and some issues may be presented in the results section. We agree that this section needs to be reorganized. Indeed, we tried to build the article with a few 'seminal' "questions" motivating the study in the scope . Some additional questions appeared in the study and the discussion of the results. In order to improve the readability of the study, we will add some 'supplementary questions' in the scope, moving the corresponding results in the results section.

3. There are too many figures. In order to reduce the number of figures in the text, some will be removed, some will be merged and some will be moved to supplementary materials.

Below we give more detailed answers to the comments made by the reviewers and make some proposals explain how we propose to modify the text if the Editor request a revised submission.

**Answer to the comments of the referee #1**

*General comments*

*This paper presents an approach for calibrating and evaluating extrapolated probabilistic hydrological predictions in the context of flood prediction. The authors consider a range of transformations for use in an uncertainty processor, and perform analysis over a large number of catchments, using multiple metrics to evaluate performance of the forecasts. The authors find that more complex transformations, which require calibration of parameters, may perform better over a calibration data-set, but typically do not perform best in an extrapolation context.*

*This is an interesting paper on an important topic, and is particularly relevant with a changing climate, where larger flooding events outside the range of historical observations may occur. The evaluation is comprehensive (large number of catchments, multiple metrics) and their analysis supports the key findings. However, I found that*

*(i) the description of the uncertainty processor, and in particular the role of the transformations, was insufficient, and*

*(ii) the discussion section requires additional work to explain the motivation for the additional analysis in this section.*

*Therefore, I recommend major revisions be made to this article before it can be published in HESS.*

As explained in the general answer above, we agree with these general comments and changes will be made accordingly. See more details below.

*Specific comments*

*More details of EHUP. The empirical hydrological uncertainty processor (EHUP) and the different transformations are described in Section 2.1.3 and 2.1.4, respectively. Unfortunately, the level of detail provided by the authors was not sufficient to allow me to understand how the EHUP works and how the transformations fit inside the EHUP. In particular, It is unclear how the transformations fit in with EHUP. A diagram or mathematical equations would help make this clearer.*

EHUP deserves indeed a more detailed description. Since the variable transformation impacts the uncertainty assessment in an extrapolation context, the role of the variable transformation within the uncertainty processor will be presented in more detail.

*It is not clear what are the inputs and outputs of the EHUP.*

We will clarify that the EHUP relies on the residuals of the discharge values available in the training data (inputs) and results in the conditional predictive distribution of the forecasted discharge.

*Why does the EHUP require a separate training period to transformation calibration period?*

The EHUP is the non parametric method that 'only' needs a training period to "build" itself, i.e. to assess the empirical residual distributions on the different variable ranges. Moreover, some of the data transformations are parametric and require a calibration data set. In order to calibrate the transformation parameters, it is necessary first to produce these empirical residual distributions. We will clarify this point in the revised version.

***Discussion section.*** *The motivation for a lot of the analysis performed in this section is not clear to me, doesn't seem to fit in with the aims of the study, and often does not seem to match the sub- headings*

*– Section 4.1*

*o Pg 25, lines 9-17: This paragraph doesn't seem to be addressing the heading of the section, which is on the number of parameters in the transformation.*

We thank the referee for pointing out that the subsection title is not appropriate. We agree. The title will be changed. We will also rephrase the 2$^{nd}$ paragraph to explain the link with the 1$^{st}$ one.

*– Section 4.2*

*o This sub-section seems like it is attempting to determine what the key drivers for performance are, but this is not evident from heading.*

Indeed, this subsection title will be rephrased as a question to describe more explicitly the section content: "What are the possible drivers for performance losses when extrapolating?". Furthermore, this question will be added to the scope as a "supplementary question" and the section will be moved to the results section accordingly.

*o Since the authors did not find any key drivers for poor performance, I'm not sure if this analysis adds much value.*

We agree with the referee on the fact that these negative results can be frustrating and do not bring much operational value. However, being able to explain when and how the performances decrease in an extrapolation context (*e.g.*, for very large and damaging floods) would be very valuable for

operational forecasters. Therefore, we think that it is important to mention that the possible 'drivers' we tested are not actual drivers. The motivation will be better explained. Furthermore, this subsection will be shortened (in particular, some figures removed or moved to the supplementary materials).

*Section 4.3*

*o The motivation for this section is unclear to me. Why are you comparing empirical and distribution based uncertainty? This seems tangential to the aims of the study. A brief sentence at the start to explain what you're looking into, and why, would be useful.*

*o You are comparing "empirical-based" and "distribution-based" uncertainty assessment in this section. Since you have not explained the EHUP in enough detail, it is not clear which of these 2 approaches you have used for the rest of this study.*

We agree with the reviewer that the motivation for this section has to be better explained. Many studies are based on methodologies combining the use of data transformations and the assumption of a Gaussian distribution [Li et al, 2017]. Morawietz et al (2011) tested this issue specifically. This is will better explained in the revised text. Furthermore, the description of the link between the variable transformation and the characterisation of the distribution (EHUP) intended in section 2.1.3 (see above) will also contribute to make the motivation clearer.

*– Section 4.4*

*o This section is about making links to previous studies, but you cite only one paper and make no comparison to the findings of that paper.*

We agree with this remark. Since there are very few papers on this issue, we do not have enough materials to carry out a full comparison with previous studies. We will remove this subsection and add a few sentences on the link to McInerney et al. (2017) in the results section and/or the conclusion section.

*– Section 5: "A need for a focus change."*

*o This section is not long enough for a separate section, and is a discussion topic. I suggest moving this to the discussion.*

We thank the reviewer and will follow his/her suggestion.

*– Limitations and future work*

*o It would be useful to have a sub-section discussing the limitations of this study and future research.*

We agree with the fact that we need to better describe the limitations and the subsequent future research. A subsection or a specific paragraph will be dedicated to the limits and perspectives.

*Too many figures. I believe this paper has too many figures. I recommend*

*– Merging some figures*

*o fig 6 and 10*

*o fig 12 and 13*

*– Is there any point in showing all 3 transformations for fig 16-18?*

*o You could consider a single transformation and combine into a single figure.*

*o Or you could move fig 17-18 to sup mat since they don't show any correlations.*

As mentioned in the general answer above, some figures will be merged or removed. Figures 12 and 13 will be merged, but we prefer to keep figures 6 and 10 separated because they are described in the text at two different places. We will move the figures 17 and 18 to the supplementary material.

> *Figure 5: I like the idea of having a diagram to explain how the different sets D1, D2sup, D2inf and D3 are used in calibration and evaluation, but I found this figure particularly confusing. In particular,*
>
> *– Why is different data used for EHUP in calibration and evaluation?*
>
> *– Why does D1 have many more points than D2sup and D3? From the text I thought D2sup and D3 had 720 points, while D1 had 500 points?*
>
> *– What's the purpose of showing the residuals on the y-axis? These are not discussed in the text.*
>
> *In panel b, most of the points for D2inf (light blue) are hidden behind points for D1 (red).*

We agree that more details are needed in the EHUP description. We will improve the description of the methodology up to subsection 2.1.4 and include more details to better understand the methods. The legend of figure 5 will be more detailed as well and we will clarify the selection of the 500 points for D1. The difference of the data used in the two steps will be better described in subsection 2.2.4.

The residuals are a key to understand the behaviour and the effects of the transformation. That is why they are discussed in section 4 (they are very important in the discussion in subsection 4.3). This will be mentioned in section 2.1.4.

> *Technical corrections*
>
> ***Abstract:*** *"... the Box-Cox transformation with a parameter between 0.1 and 0.3 can be a reasonable choice for flood forecasting"*
>
> *You have only shown results for lambda=0.2 in this paper. How you can say that using lambda between 0.1 and 0.3 can be a reasonable choice?*

We agree that this result is not shown in the submitted version: as explained in the methodology section, we studied a large number (17) of parameter values but we did not show the results for all of them, for the sake of brevity. The Box-Cox transformation has a "smooth" effect with respect to λ. We will add a figure in supplementary material.

> ***Table 1:*** *Change "percentiles" to "quantiles"*

This word will be changed.

> ***Pg 5, line 10:*** *"For each catchment, the lag time LT is estimated as the lag time maximising the cross-correlation between rainfall and discharge time series."*
>
> *What is the relevance of estimating LT? This becomes more obvious later in the paper, but should be described briefly here.*

Lag time is relevant to describe the catchment behaviour in a forecasting purpose: this characteristic duration has to be compared to the lead time (a) for the data assimilation procedures (most operational forecasting models use some) and (b) the relative importances of observed and forecasted precipitation inputs (which can explain part of the predictive performance when real precipitation forecasts are used). This will be better explained.

*Pg 5, line 15:* *"It is a deterministic lumped storage-type model that uses catchment areal rainfall and PE as inputs"*

*What rainfall is used to produce the GRP forecasts? Is observed rainfall used, forecast rainfall, etc? If it is observed rainfall, then how is this used in a forecasting context?*

We used the framework designed by Krzysztofowicz *et al.* in various studies, which separates the input uncertainty (mainly the observed and forecasted rainfall) and the hydrological uncertainty. This study focuses only on the 'effect' of the extrapolation degree in the hydrological uncertainty when using the best available rainfall product. In a forecasting context, when using uncertain rainfall, we will combine input uncertainty (rainfall) and hydrological uncertainty, as done for example in Bourgin et al. (2014).

*Pg 6, line 3-4:* *"Since herein only the ability of the post-processor to extrapolate uncertainty quantification is studied, the model is calibrated in forecasting mode over the 10-year series by minimising the sum of squared errors for a lead time taken as the lag time LT."*

*What is meant by "forecasting mode" here? More details on how forecasts are generated would be useful.*

The "forecasting mode" is to be compared to the "simulation mode" where no data assimilation is used. The latter allows to test the simulation model alone and assess its 'own' performance. The former is used to test a model in a context which is closer to the operational context (of the Flood Forecasting Service). Some references and a reference to appendix (where this is explained) will be added.

*Pg 7, lines 8-12:* *Is the NQT actually used in this study? If so, it's not clear how and where it's used.*

*Pg 7, lines 14-15:* *If the NQT requires additional assumptions for the tails, how do you handle this problem in in this study?*

We thank the referee for pointing out that this point is unclear. NQT was not tested, mainly because this transformation is known to require a particular care in an extrapolation context (see the technical note by Bogner *et al.* (2012) who explained that additional assumptions have to be made). However, since it is a frequently used transformation, we think that it is relevant in the introduction section. We will move this description at the very end of the subsection and explain why it was not used.

*Pg 7, lines 30-31:* *"McInerney et al. (2017) obtained their best results with lambda = 0.2 over 17 perennial catchments."*

*What do you mean by "best results"? Please provide some context for this statement.*

We used the paradigm set by Gneiting *et al.* (2007): the results are the "best" in terms of (1) reliability and (2) sharpness.

*Pg 8, line 3:* *Why does this equation use different notation than other transformations?*

We thank the referee for pointing this inconsistency, which could be confusing for the reader. The notation will be made homogeneous.

*Pg 10, line 6:* *"maximum discharge of time series"*

*Make it clear you are referring to forecast discharge here.*

Changes will be made accordingly to this suggestion.

*Pg 10, line 8: "the first time step"*

*What do you mean by "first time step"? Do you mean the closest time step?*

215 We will better explain that the first time step of the event is the closest time step preceding the peak time step such as all discharge values from this time step to the peak are larger than 20 % (25%) of the peak flow value.

*Pg 10, lines 20-21, Pg 11, line 1: The purpose of the "control", "training", and "calibration" subsets has not been explained. Please describe what they are used for.*

220 A short paragraph will be added to explain why the use of a variable transformation within an empirical HUP requires to use three subsets to test the performances in an extrapolation context.

*Pg 12, line 1: It is unclear what the "coverage rate of the 80% predictive intervals" is. Please provide equations or description.*

We will add that these '80% predictive intervals' are bounded by the 0.1 and 0.9 quantiles of the
225 predictive distributions.

*Pg 13, line 6: "i.e. from the distribution of the observed discharges over the events selected"*

*What is meant by "events selected"? Is this all events in G1, G2 and G3?*

We will clarify that the "events selected" refer to the events in the data subset for calibration or
230 control.

*Pg 15, lines 6-8: "as expected, and that there is no significant difference between the calibrated Box-Cox transformation (d), the calibrated log-sinh transformation (e) and the best performing transformation (f)."*

*How are you determining whether differences between results are "significant"? A
235 statistical test should be used to determine whether differences are "significant".*

A Mann-Whitney test has been used. It showed no significant difference between the reliability criterion values distributions obtained with the calibrated transformations. However, what we meant here is mainly that no difference can be noticed from Fig. 6. This paragraph will be rephrased in order to refer to what can be inferred from Fig. 6.

240 *Pg 15, line 8: Is "best performing" the same as "best calibrated"? If so, use a single term.*

We thank the reviewer for pointing out that this difference could be somewhat confusing. A single expression will be used.

*Pg 15, lines 9-10: "Interestingly, the log transformation provides the best results for
245 the other criteria (not used as the objective function)."*

*Are these results shown anywhere? If so, provide reference to figure.*

This sentence will be removed (see the answer to the second referee).

*Pg 15, line 13-14: "While the log-transformation behaviour is frequently chosen for LT/2 and LT, the additive behaviour becomes more frequent for 2 LT and 3 LT."*

250 *It is unclear what you mean by "additive behaviour" and how this is seen in the figures (i.e. what parameters relate to additive behaviour).*

The additive behaviour refers to the behaviour of the no transformation. A link to subsection 2.1.4. (page 7) where this is detailed, will be added.

> **Pg 17, lines 7-8:** *"This confirms that the CRPSS itself is not sufficient to evaluate the adequacy of uncertainty estimation"*
>
> *Similar findings about CRPS being insensitive to chosen data transformation have been made in other studies, e.g. Woldemeskel et al. (2018). It might be worth mentioning this.*

We did not know this article. Thank you for giving this reference. We will mention it in the revised article.

> **Figure 9 caption:** *"Thérain River at Beauvais (755 km2): the forecasts are reliable and ...."*
>
> *This statement does not seem correct. I would say the forecasts are not reliable for "none".*

This comment is very true, we implicitly described only the uncertainty assessment when a variable transformation is used, since such a transformation is most often needed to achieve (more or less) reliable results. "(except if no transformation is used)" will be added.

> **Figure 14:** *Legend is missing*

We apologize for this missing legend. Legend is given below the figure.

> **Pg 25, line 22-23:** *"The results indicate that it is not possible to anticipate the alpha-index values when extrapolating high flows in D3 based on the alpha-index values obtained when extrapolating high flows in D2sup."*
>
> *There appears to be some correlation in Figure 16. What is the Spearmen correlation?*
>
> **Pg 25, lines 27-28:** *"In both cases, no trend appears, regardless of the variable transformation used, with Spearman correlation coefficients lower than 0.5."*
>
> *A Spearman correlation of 0.5 does not seem correct. If it was 0.5, then there would be a clear trend.*

Spearman values were all lower than 0.33 .

> **Pg 25, line 31:** *What is a "normalized RMSE" and why is it used? A sentence/equation describing this would be useful (rather than just a citation).*

We thank the referee for having detected the absence of description of this criterion. A description will be added in section 2.3.1.

> **Pg 26, line 11:** *"Even if there is no theoretical advantage to using the Gaussian distribution calibrated on the transformed-variable residuals rather than the empirical distribution to assess the predictive uncertainty, we tested the impact of this choice."*
>
> *If there is no theoretical advantage, why are you testing this?*

As said in the general answer, the motivation of the subsection 4.3 will be better explained at its beginning.

> **Pg 33, line 21:** *"the Box-Cox transformation with its lambda parameter set at 0.2 or between 0.1 and 0.3."*
>
> *You have only shown results for lambda=0.2 in this paper. How you can recommend using other values of lambda between 0.1 and 0.3 in the conclusions of this paper?*

As explained above, this result was indeed not shown in the submitted version for the sake of brevity but we studied a large number (17) of parameter values. We will add a figure in the supplementary material.

> *Section B2.2. This relationship was shown by McInerney et al. (2017) (Appendix A)*

We agree that this relationship was also pointed out by McInerney *et al.* (2017). This will be acknowledged in the text.

> *References*
>
> *McInerney, D., Thyer, M., Kavetski, D., Lerat, J. & Kuczera, G. 2017. Improving probabilistic prediction of daily streamflow by identifying Pareto optimal approaches for modeling heteroscedastic residual errors. Water Resources Research, 53.*
>
> *Woldemeskel, F., McInerney, D., Lerat, J., Thyer, M., Kavetski, D., Shin, D., Tuteja, N. & Kuczera, G.2018. Evaluating residual error approaches for post-processing monthly and seasonal streamflow forecasts. Hydrol. Earth Syst. Sci. Discuss., 2018, 1-40.*

**Answer to the comments of Dr. Engeland (referee #2)**

> *The paper presents a framework aiming at evaluating the performance of probabilistic forecasts on highest flood events that the post-processors are not calibrated for. The authors combine an empirical hydrological post-processor (EHUP) with different trans-formations, and compare the performance of the predictive distributions for forecasted floods that are higher than the floods used for training/calibrating the EHUP and the transformations.*
>
> *The paper is interesting and deserves publication following a major revision. Below It lists some important issues to be addressed in the revised manuscript.*
>
> *Throughout the introduction, the importance of modelling the heteroscedasticity of the predictive uncertainty distribution emphasized. I miss a good argument why it is important (i.e. to obtain reliability), and you could refer to literature that shows this (i.e. McInerney et al., 2017). In the introduction and discussion, you ignore that other properties of the predictive distribution (i.e. bias and skewness) could also depend on forecasted flows. My experience is that a calibrated hydrological model tends to under-estimate flood peaks, introducing a possible bias. Bremnes et al (2019) shows that the skewness depends on the predicted wind and that adding this property improves the forecasts for high wind speeds. You discuss this briefly in lines 4-9 on page 8. Is it possible that the results presented in Figures 12 and 13 indicate that the skewness is an important issue for the reliability of the predictive distributions, and that your approach has to small skewness?*

The referee points out here an important fact. The heteroscedasticity is an important property to describe a probability distribution and is often looked out in the literature, but it is very true that all the properties of the distribution have to be checked. This issue will be mentioned in the introduction and the conclusion.

We are not sure that figures 12 & 13 give any indication on the skewness. They only describe the reliability of two predictive quantiles. They show that the evolution in an extrapolation context of the empirical distribution assessed by EHUP is not perfectly reliable.

Indeed, the main issue here is the stability of the overall predictive distribution in an extrapolation context. The tests provided in section 4.3 give some insights.

*I miss an explanation of which meteorological products you used to generate the discharge forecasts. Did you use the reanalysis mentioned in 2.1.1 or did you use a forecast product?*

340 We used the reanalysis mentioned in 2.1.1 as meteorological inputs. We chose to follow the decomposition proposed by Krzysztofowicz (input uncertainty and modelling uncertainty): here we test only the modelling uncertainty in extrapolation. Further work shall investigate the contribution of the input uncertainty (Bourgin, 2014) in an extrapolation context. This will be mentioned in section 2.1.1 and in the conclusion.

345 *The EHUP needs a better description, in particular how it is used in combination with the different transformations. I also need a clarification of which data were used for estimating the empirical quantiles of errors. On page 6 you write that the top 5% pairs ranked by simulated values are used, whereas on page 11 you write that the subsets D1 and D2 were used. Figure 5 indicates that not the whole D1 subset was used for*
350 *training of the EHUP, only the highest discharge values. A consistent description is needed to avoid confusion.*

We thank both reviewers and agree with them on the fact that EHUP needs a better description. It will be done following their comments. Note that the 5%-selection is made on the subset used for the training (D1 for the calibration step and D1 + D2 for the evaluation step). Figure 5 and its
355 legend will be improved to make clear that only the top 5% pairs are used for the extrapolation.

*The discussion section needs a better organization. Results presented in section 4.1could be integrated into section 3. In Figure 15, the only new result is the boxes labelled 'g'. Could it be integrated into Figure 10? Section 4.2 and 4.3 introduces new results that do not directly relate to the objectives / questions listed on Page 4. If these*
360 *results should be included, you could add one more objective related to these results, and integrate the results into Section 3. I suggest to exclude results and discussion in section 4.3 (including Figure 19 and 20) and only briefly summarize the main findings.*

As mentioned in the general answer above, we agree and look forward a better organization of the section. We prefer to keep subsections 4.1 and 4.3, because they mostly bring information to
365 interpret the main results. In order to do so, we achieved a few complementary tests. The issue in subsection 4.3 seems particularly important because this assumption is often made but sometimes not tested. The scope (subsection 1.3) will be completed in order to make it appear at the beginning of the article. We will place the 2nd figure in the supplementary materials.

*Section 5 could be also a part of the discussion.*

370 We agree. Section 5 will be included as the last subsection in the discussion.

*The number of figures could be reduced. Figure 2 – right panel is not necessary.*

We agree that both panels of figure 2 are not necessary, but we prefer keeping the right panel because it better explains the effect of the transformation on the uncertainty assessment: a constant probability distribution in the transformed space will evolve in the untransformed space based on
375 the behaviour of the inverse data transformation.

*Figures 4a and 4b could be merged. Is it possible the merge Figure 5a and b? Could result sin Figure 15 be included in Figure 10? Figure 11 is not necessary. Figure 12, and 13could be merged. I suggest to remove Figure 14a since it is just another measure of reliability and does not add new information to the results. Figures 19 and*
380 *20 could be excluded or moved to supplementary materials.*

We thank the referee for pointing out that some figures can be rearranged or merged. Figures 4a and 4b will be merged. However we did not manage to merge figures 5a and 5b in a unique meaningful and easy-to-read figure. Results in figure 15 will be added to figure 10. We respectfully disagree on figure 11, which we consider interesting since it is the only one displaying a scatter plot (while most of the figures display box-plots), which brings an additional and valuable information: the comparison catchment per catchment. Figures 12 and 13 will be merged. Figure 14a provides indeed another reliability criterion but this one brings another information (both α-index and coverage ratio criteria are synthetic criteria) and is important for many operational forecasters.

> *Below follows some detailed comments to the manuscript:*
>
> *Table 2: When you compare discharge across catchments, I think it is better to use specific discharge (l/s/km²).*

We agree. Peak discharges will be describe through specific discharge values.

> *Figure 3: What is the explanations for this apparently negative skewness for the predictive distribution? The log-transformation leads to slightly positively skewed predictive distribution?*

The empirical distribution provided by EHUP reflects the assessed distribution on the training data set. The log transformation exacerbates the skewness, since it has a "multiplicative effect".

> *Figure 14: Legend is missing*

We apologize for the missing legend. Legend is given below the figure.

> *Page 2: The meaning of the first paragraph of section 1.2 is difficult to understand. In particular the two first sentences needs more context.*

We thank the referee for this warning. The paragraph will be rewritten, giving the context of operational forecast systems and organization, in order to provide useful information to crisis managers.

> *Page 3: I suggest to write the first paragraph of 1.2.1 as: "A first approach that intends to model each source of uncertainty separately and to propagate these uncertainties through the modelling chain is presented in Renard et al., (2010). According to this approach, the heteroscedasticity of the predictive uncertainty distribution results from the separate modelling of each source of uncertainty and from the statistical model specification. While this approach is promising, operational application can be hindered by the challenge of making the hydrological modelling uncertainty explicit, as pointed out by Salamon and Feyen (2009)."*

We thank the referee and adopt his proposal.

> *Question or the paragraph above: which statistical model needs to be specified? Is it for the meteorological input or for the simulated discharge?*

Renard et al. (2010) use a Bayesian modelling, which needs a full specification of the inputs distribution (assumptions) and of the likelihood (another assumption).

> *Page 4 lines 7-8: These approaches are not exclusive of each other. Even when future precipitation is the main source of uncertainty, post processing is often required to produce reliable hydrological ensembles Question: What does 'these approaches' refer to? does it refer to all approaches presented in the introduction or all approaches presented in section 1.2.2?*

We agree that this sentence is not clear. "These approaches" refer to the two main families described in subsections 1.2.1 and 1.2.2. This will be specified in the revised manuscript.

> *Page 5 Section 2.1.1: Maybe a question of style, you write 'We used a set of 154 unregulated catchments spread throughout France (Fig. 1) to test our hypotheses over various hydrological regimes and forecasting contexts." Since you have chosen to use formulate research questions and not to test hypotheses in this paper, the sentence could be changed to 'We used a set of 154 unregulated catchments spread throughout France (Fig. 1) over various hydrological regimes and forecasting contexts to provide robust answers to our research questions.*

We agree and will change the text accordingly.

> *Page 7, line 19: You write that the log-transformation is non-parametric. I would rather say it is a parametric transformation with no tuning parameters. The term non-parametric is often used when you make no assumptions about the form or parameters of the transformation.*

We agree that the term "non-parametric" is frequently used for distributions and means that there is no assumption about the form of the distribution. This word can also be used for transformations of functions. Then it only refers to the existence of tuned parameters. We will clarify the meaning in the text.

> *Page 10 Section 2.2.2. How did you select more than one event? According to the description you selected one event defined by the maximum discharge of the time series.*

Once the first event is selected, the process is iterated over the remaining data to select more events. This point will be detailed in the revised text.

> *Page 11: Why has the calibration data subset to encompass time steps with simulated discharge values higher than those of the training subset?*

Since our intention is to test the robustness and adequacy of different data transformations in an extrapolation context, it is more useful to calibrate their parameters in an extrapolation context, i.e. on simulated discharge values larger than the ones met in the training step. In addition, since we used an empirical uncertainty processor, the data transformations have almost no impact on the uncertainty estimation in the training subset and we will not be able to "tune" their parameters.

> *Page 13: First equation: define k and N Second equation: Could you use the same notation as in the first equation. i.e. write it as sum divided by number of time steps?*

N is the number of time steps on which the CRPS is computed and k is just an index. We will precise the meaning of N and write the second equation using the same notation.

> *Page 15, lines 9-10: Here you comment results that are not yet presented, making it difficult for the reader to follow. I think this sentence fits better in the discussion*

We thank the referee for his careful review. This sentence corresponds to some results that were not shown. It will be removed in the revised manuscript.

> *Page 16: The last three lines have to be re-phrased in order to make sense: "In operational settings, non-exceedance frequencies of the lower (0.1 quantile) and the exceedance frequencies of the upper (0.9 quantile) bounds of the predictive distribution are of particular interest. It is expected that those values remain close to 10% for a reliable predictive distribution. Deviations from these frequencies indicates biases in the estimated quantiles."*

We thank the referee for his proposal. The sentences will be rewritten.

> *Page 17 lines 3-5: I think it is better to write something like this (I think it is better to write that the 0.1 and 0.9 quantiles are over or under-estimated, and not the (non)-exceedance frequency of the (0.1) and 0.9 quantiles.): "More importantly, it can be seen that the lack of reliability of the log transformation seen for 3 LT in Fig.10 appears to be related to an underestimation of both the 0.1 and 0.9 quantile. Compared to the other transformations, the log transformation has the largest under-estimation of the0.1 quantile and the smallest under-estimation of the 0.9 quantile."*

The sentences will be rewritten to make them clearer.

> *Page 18 Section 3.2.2: Please be more precise in the comments: What is 'overall performance' ? Suggestion for re-phrasing some of the sentences:"We note that the log transformation has the highest median value for the coverage ratio, and is also the closest to the 80% ratio that is expected from a reliable forecast,""In addition, the CRPSS and the NSE distributions have limited sensitivity to the variable transformation. We can even see that not using any transformation yields slightly better results according to NSE.*

The "overall performance" refers to an "overall" criterion which does not investigate a specific property of the forecasts (reliability, accuracy, sharpness…) but intends to describe the whole predictive distribution. We used the CRPS, as mentioned in subsection 2.3.1. It will be written in section 3.2.2 as well to make it clearer.

> *Page 33: Please provide clear conclusions related to each of the objectives and answer the research questions asked in the introduction.*

We thank the referee for this suggestion that we will follow in the revised version of this article.

> *New reference in this review: Bremnes, J.B., 2019: Constrained Quantile Regression Splines for Ensemble Post processing. Mon. Wea. Rev., 147, 1769–1780,https://doi.org/10.1175/MWR-D-18-0420.1*

| Line in the answer to referees' comments (*) | Page in the first submitted version | Line in the first submitted version | Changes made in the second submitted version |
|---|---|---|---|
| 328 | 1 | 8 | The sentence has been completed as:

"*to account for heteroscedasticity and the evolution of the other properties of the predictive distribution with the discharge magnitude.*" |
| 402 | 2 | 21 | The beginning of the subsection 1.2 has been rewritten to better refer to the context of operational services:

"*Even if significant progress has been made and implemented in operational flood forecasting systems (e.g., Bennett et al., 2014; Demargne et al., 2014; Pagano et al., 2014), some uncertainty remains. In order to achieve an efficient crisis management and decision making, communication of reliable predictive uncertainty information is therefore a prerequisite (Todini, 2004; Pappenberger and Beven, 2006; Demeritt et al., 2007; Verkade and Werner, 2011). Hereafter, reliability is defined as […]*". |
| 328 | 2 | 30/31 | The sentence has been rewritten as:

"*In an extrapolation context, it is of utter importance that the predictive uncertainty assessment provides a correction description of the evolution of the predictive distribution properties with the discharge magnitude. Bremnes (2019) showed that the skewness of wind speed distribution depends on the forecasted wind. Modelled residuals of discharge forecasts often exhibit high heteroscedasticy (Yang et al., 2007). McInerney et al. (2017) focused their study on representing error heteroscedasticity of discharge forecasts with respect to simulated streamflow. To achieve reliable forecasts, a correct description of the heteroscedasticity, either explicitly or implicitly, is necessary.* |
| 413 | 3 | 6 | The first paragraph of the subsection1.2.1 has been rewritten as:

"*A first approach that intends to model each source of uncertainty separately and to propagate these uncertainties through the modelling chain is presented in Renard et al., (2010). According to this approach, the heteroscedasticity of the predictive uncertainty distribution results from the separate modelling of each source of uncertainty and from the statistical model specification. While this approach is promising, operational application can be hindered by the challenge of making the hydrological modelling uncertainty explicit, as pointed out by Salamon and Feyen (2009).*" |
| 328 | 4 | 13 | The sentence has been rewritten as:

"*Note that many of these approaches use a variable transformation to handle the heteroscedasticity (and more generally the evolution of the predictive distribution properties with respect to the forecasted discharge).*" |

| Line in the answer to referees' comments (*) | Page | Line | Changes made in the second submitted version |
|---|---|---|---|
| | | **in the first submitted version** | |
| | 4 | | The last sentence of the first paragraph of the Scope subsection has been rewritten as: "*Since achieving a reliable predictive uncertainty assessment in an extrapolation context is a challenging task likely to remain imperfect if the stability of the characteristics of the predictive distributions is not properly ensured, it requires a specific crash-testing framework~\citep{Andreassian2009}. The objectives of this article are:*" |
| 79 & 363 | 4 | 28 | The "Scope" subsection has been completed with a 3$^{rd}$ question:

 « *We attempt to answer three questions : (a) Can we improve […] ? (b) Do more flexible transformations […] ? (c) If there is a performance loss when extrapolating, is there any driver that can help the operational forecasters to predict this performance loss and question the quality of the forecasts ?* » |
| 423 | 4 | ### | A new subsection has been set up to more clearly 'separate' the two last paragraphs and the sentence has been rewritten as :

 "*The approaches presented in subsections 1.2.1 and 1.2.2 are not exclusive of each other.*" |
| 432 | 5 | 4 | The sentence has been rewritten as:

 "*We used a set of 154 unregulated catchments spread throughout France (Fig. 1) over various hydrological regimes and forecasting contexts to provide robust answers to our research questions.*" |
| 166 | 5 | 10 | A sentence has been added to point out the importance of LT and to test different lead times:

 "*When a hydrological model is used to issue forecasts, it is often necessary to compare the lead-time to a characteristic time of the catchment (section 2.1.2). For each catchment [...]*" |
| 187 | 5 | 15 | The last sentence has been completed as:

 "*In forecasting mode (appendix A), the model also assimilates [...]*" |
| 161 | 5 | | The word "*Percentiles*" have been replaced by the word "*Quantiles*" in Tab. 1 |
| 166 | 6 | 6 | A reference to Berthet *et al.* (2009) has been added. |
| **46 & 54 & 352** | 6 | 7 | The description of the EHUP (mainly in subsection 2.1.3.) has been modified in deep, so that it brings information enough to make the methodology clear. |
| 54 | | | The role of the residuals and of the inverse transformation have been emphasized. |

| Line in the answer to referees' comments (*) | Page | Line | Changes made in the second submitted version |
|---|---|---|---|
| | | in the first submitted version | |
| 143 & 352 | | | The meanings of the selection of the 500 top values in D1 and the role of the 5%-highest values have been clarified.

The legend of Fig. 5 has been rewritten to make clearer that only the information conveyed by the top 5% pairs of the training data subset are used by EHUP to extrapolate on the calibration or control periods. |
| 175 & 340 | 6 | 4/5 | A sentence is added after the sentence whose 1$^{st}$ words are "*Since herein only the ability of the post-processor to extrapolate uncertainty quantification is studied*":

"*For the same reason, the model is fed only with observed rainfall (no forecast of precipitation), in order to reduce the impact of the input uncertainty.*" |
| 196 | 7 | 8 | The NQT description has been sent at the end of the subsection 2.1.4 and starts with:

"*Another common variable transformation is the normal quantile transformation (NQT). It is a [...]*"

The following sentence has been added at the end of the paragraph:

"*This is why we did not test this transformation in this study focused on the extrapolation context.*" |
| 196 | 7 | 16 | The sentence has been changed as:

« *Three analytical transformations are often met in hydrological studies: [...].* » |
| 437 | 7 | 19 | The following words have been added at the end of the sentence "*This transformation is then non-parametric*":

"*(no parameter has to be calibrated).*" |
| 205 | 7 | 31 | The sentence has been changed as:

« *McInerney* et al. *(2017) obtained their most reliable and sharpest results with [...]* » |
| 208 | 8 | 3 | The equations formalism has been made homogeneous. |

| Line in the answer to referees' comments (*) | Page | Line | Changes made in the second submitted version |
| | in the first submitted version | | |
| --- | --- | --- | --- |
| 372 | 8 | | The left panel of the figure 2 has been removed: only the right panel is kept. A sentence has been added to its legend:

"*the inverse transformations are displayed because they explain the final effect of the transformation on the uncertainty assessment: the constant probability distribution in the transformed space (provided by the EHUP) will result in an distribution in the untransformed space, whose evolution depends on the behaviour of the inverse data transformation.*" |
| 212 | 10 | 6 | The fact the selection is made on the forecasted discharge series has been clarified:

« *(1) the maximum forecasted discharge of the time series was selected [...]* » |
| 215 | 10 | 7 | The words "*the first time step*" have been replaced by "*the preceding (following) time step closest to the peak*" |
| 444 | 10 | 10 | The following sentence has been added before the sentence which starts as "*A minimum time lapse of 24 h was enforced between two events […]*":

"*The process is then iterated over the remaining data to select all events.*" |
| 392 | 10 | Tab. 2 | The median value of the peak specific discharges are given (instead of peak discharges) |
| 143 | 11 | 22 | The difference between the data used for the training in the calibration step and in the evaluation step has been clarified in order to better explain the objective of the methodology:

« *In the second step, the EHUP was trained on a data set which encompassed D1, D2inf and D2sup using the parameter set obtained during the calibration step. Then, the predictive uncertainty distribution was evaluated on the control data set D3. Training the EHUP on the union of D1, D2inf and D2sup allows to control the uncertainty assessment from small to large degrees of extrapolation (on D3). Indeed if we had kept the training on D1 only, we would have not been able to test small degrees of extrapolation on independent data for every catchment (see the discussion in Sect. 3.3).* » |
| 448 | 11 | 23 | The following sentences have been added before the sentence "*The parameter set obtaining the best criterion value was selected.*":

"*Indeed, the data transformations have almost no impact on the uncertainty estimation by EHUP on events of the same magnitude as those of the training subset. Therefore the calibration subset has to encompass events of a larger magnitude (D2Sup).*" |

(*) In *190803_Answers to revisions.pdf*. Before line 307 (blue): 1st (anonymous) referee ; after line 30 (green) : 2nd referee (Dr. Engeland)

| Line in the answer to referees' comments (*) | Page | Line | Changes made in the second submitted version |
|---|---|---|---|
| | | in the first submitted version | |
| 224 | 12 | 1 | The sentence has been completed as:

« *[…] the coverage rate of the 80 % predictive intervals (bounded by the 0.1 and 0.9 quantiles of the predictive distributions […]* » |
| 381 | 12 | | Fig. 4a and 4b have been merged.

Furthermore, the legend has been modified: "*Illustration of the selection of the data subsets for the Ill River at Didenheim (668 km2 ). First, the events are selected (grey highlighting). Then, the four data subsets are populated according to the thresholds (horizontal dashed lines). See Sect. 2.1.3 for more details.*" |
| 455 | 13 | 4 | The sentence has been modified as "*where N is the number of time steps, F the predictive cumulative distribution, H the Heaviside function and […]*" |
| 229 | 13 | 6 | The words "*over the events selected*" have been replaced by "*over the same data subset*". |
| 455 | 13 | 9 | The second equation has been rewritten using the same notation.

$$\frac{\sum_{k=1}^{N} q_{0.9}(Q_k) - q_{0.1}(Q_k)}{\sum_{k=1}^{N} Q_{k,obs}}$$    "*Where $q_{0.9}(Q_k)$ is the quantile 0.9 of the predictive distribution at time step k.*" |
| 143 | 14 | | Figure 5. has been modified to better emphasize the role of the top 5%-forecasted discharge values. |
| 236 | 15 | 7 | The words "*there is no significant difference*" have been replaced by "*no noticeable difference can be seen in Fig. 6 between […].*" |
| 242 | 15 | 8 | The words "*the best performing transformation (f)*" have been replaced by "*the best calibrated transformation (f)*". |
| 247 & 459 | 15 | 9 | The sentence "*Interestingly, the log transformation provides the best results for the other criteria (not used as the objective function.*" has been removed. |
| 252 | 15 | 14 | The words "(corresponding to the use of no transformation, see *Sect. 2.1.4)*" have been added after "*additive behaviour*". |

| Line in the answer to referees' comments (*) | Page | Line | Changes made in the second submitted version |
| --- | --- | --- | --- |
| | | in the first submitted version | |
| 467 | 16 | 17/19 | The sentence starting as "*In operational settings, non-exceedance […]*" has been rewritten as:

 "*In operational settings, non-exceedance frequencies of the quantiles of the predictive distribution which are the lower and upper bounds of the predictive interval communicated to the authorities are of particular interest. The 80%-predictive interval (bounded by the 0.1 and 0.9 quantiles) is often used. It is expected that the non-exceedance frequencies of the lower bound and the exceedance frequencies of the upper bound remain close to 10% for a reliable predictive distribution. Deviations from these frequencies indicates biases in the estimated quantiles.*" |
| 475 | 17 | ### | The sentence has been rewritten as:

 "*More importantly, it can be seen that the lack of reliability of the log transformation for the 3-LT lead time seen in Fig. 10 appears to be related to an underestimation of the 0.1 quantile which is more important than for the other tested transformations, while the 0.9 quantile is less underestimated than for the other transformations.*" |
| 483 | 18 | 2 | The words "*(measured by the CRPSS)*" have been added after the words "*namely the overall performance*" |
| 259 | 18 | 6 | The sentence is completed and becomes « *In addition, the CRPSS and the NSE distributions have limited sensitivity to the variable transformations (also shown by Woldemeskel et al., 2018, for the CRPS), even if […]* » |
| 265 | 19 | | In the legend of Fig. 9, the words "*(except if no transformation is used)*" are added at the end of the comment for the Thérain River at Beauvais. |
| 381 | 20 | | Fig. 10 & 15 have been merged. |
| 269 & 399 | 24 | | Legend of Fig. 14 is given below the figure. |
| 74 | 25 | 2 | The title of the 1st subsection of the discussion has been changed as:

 "*Do more complex parametric transformations yield better results in an extrapolation context ?*" |
| 74 | 25 | 9 | The first sentence of the 2nd paragraph of subsection 4.1 has been rewritten as:

 "*These results could be explained by the fact that the calibration did not result in the optimally relevant parameter set. To investigate whether [...]*" |

| Line in the answer to referees' comments (*) | Page | Line | Changes made in the second submitted version |
|---|---|---|---|
| | | in the first submitted version | |
| 79 | 25 | 18 | The title of the 2nd subsection of the discussion has been changed as: "*Investigating the performance loss in an extrapolation context*" |
| 79 & 363 | 25 | 18 | The subsection 4.2 has been moved to the results section (and becomes the subsection 3.3). |
| 85 | 25 | 18 | A sentence has been added to start the subsection dedicated ot the performance loss in an extrapolation context (initially subsection 4.2, now subsection 3.3) "*It is very important that operational forecasters can predict when they can trust in the forecasts issued by their models and when their quality becomes questionable. Therefore we investigated […]*" |
| 278 | 25 | 27/28 | The end of the sentence has been modified in : "*with Spearman coefficients values (much) lower than 0.33*" |
| 100 | 26 | 1 | The title of the 3rd subsection of the discussion has been changed as: "*Empirical-based versus distribution-based approaches : does the distribution shape choice impact the uncertainty assessment in an extrapolation context?*" |
| 100 & 287 | 26 | 2 | In order to better explain the objectives and motivations of this analysis, the subsection now starts by these sentences: « *Besides the reduction of heteroscedasticity, many studies use post-processors which are explicitly based on the assumption of a Gaussian distribution and use data transformations to fulfil this hypothesis [Li et al, 2017]. Some post-processors are based on it, such as the MCP or the meta-Gaussian model, and the NQT was designed to precisely achieve it. Morawietz et al (2011) tested this issue. We first checked whether the variable transformation helped to reach a Gaussian distributed of the residuals computed with the transformed variables. Then we investigate whether better performance can be achieved using eimpirical transformed residuals distributions or using Gaussian distributions calibrated on these empirical distributions.* We used the Shapiro-Francia test. [...]* » |
| 109 | 29 | 1 | The subsection 4.4 has been removed. The links with the research by McInerney *et al.* (2017) have been recalled in the results and conclusion sections. |
| 363 | 30 | | Fig. 20 has been placed in Supplementary materials |

| Line in the answer to referees' comments (*) | Page | Line | Changes made in the second submitted version |
|---|---|---|---|
| | | in the first submitted version | |
| 116 & 370 | 32 | 1 | The section 5 has been included as the last subsection of the discussion. |
| 328 | 33 | 9/10 | The sentence has been rewritten as:

 "*The latter has to handle the heteroscedasticity and the evolution of the other predictive uncertainty distribution properties, which is very problematic in an extrapolation context to issue reliable uncertainty assessment.*" |
| 489 | 33 | 12/… | The conclusion has been reorganized. It now includes a "*Main findings*" subsection and a "*Limitations and perspectives*" subsection. In the former, an answer is explicitly provided to every research question specified in the scope subsection (a), (b) and (c). |
| 293 | 33 | 21 | *A figure is provided in supplementary materials to show it.* |
| 85 & 120 & 340 | 33 | 25 | The subsection "*Limitations and perspectives*" of the conclusion includes:

 " *We used the framework designed by Krzysztofowicz et al. in various studies, which separates the input uncertainty (mainly the observed and forecasted rainfall) and the hydrological uncertainty. This study focuses only on the 'effect' of the extrapolation degree in the hydrological uncertainty when using the best available rainfall product. Future works should combine both input uncertainty (rainfall) and hydrological uncertainty (Bourgin et al., 2014), to evaluate the impact of using uncertain (forecasted) rainfall in a forecasting context.*

 *We found no variable correlated to the performance loss we observed in an extrapolation context. Testing more variables potentially correlated is necessary. First it may open new perspectives to explain these losses and improve our understanding of the flaws of the hydrological model and of the EHUP. Furthermore, it would be very useful to help the operational forecasters to detect major events when their forecasts have to be particularly questioned.*

 *Furthermore, improving the regionalisation of the predictive distribution assessment, as proposed in Bourgin et al. (2015) and Bock et al. (2018) could help build more robust assessment of uncertainty quantification when forecasting high flows.*" |
| 297 | 37 | 13 | The sentence has been changed in "*As pointed out by McInerney* et al. *(2017), when alpha << y […]*" |
| 125 & 381 | 22 & 23 | | Fig. 12 & 13 have been merged |

(*) In *190803_Answers to revisions.pdf*. Before line 307 (blue): 1st (anonymous) referee ; after line 30 (green) : 2nd referee (Dr. Engeland)

| Line in the answer to referees' comments (*) | Page | Line | Changes made in the second submitted version |
|---|---|---|---|
| | | in the first submitted version | |
| 116 & 370 | 32 | 1 | The section 5 has been included as the last subsection of the discussion. |
| 328 | 33 | 9/10 | The sentence has been rewritten as:

 "*The latter has to handle the heteroscedasticity and the evolution of the other predictive uncertainty distribution properties, which is very problematic in an extrapolation context to issue reliable uncertainty assessment.*" |
| 489 | 33 | 12/… | The conclusion has been reorganized. It now includes a "*Main findings*" subsection and a "*Limitations and perspectives*" subsection. In the former, an answer is explicitly provided to every research question specified in the scope subsection (a), (b) and (c). |
| 293 | 33 | 21 | *A figure is provided in supplementary materials to show it.* |
| 85 & 120 & 340 | 33 | 25 | The subsection "*Limitations and perspectives*" of the conclusion includes:

 " *We used the framework designed by Krzysztofowicz et al. in various studies, which separates the input uncertainty (mainly the observed and forecasted rainfall) and the hydrological uncertainty. This study focuses only on the 'effect' of the extrapolation degree in the hydrological uncertainty when using the best available rainfall product. Future works should combine both input uncertainty (rainfall) and hydrological uncertainty (Bourgin et al., 2014), to evaluate the impact of using uncertain (forecasted) rainfall in a forecasting context.*

 *We found no variable correlated to the performance loss we observed in an extrapolation context. Testing more variables potentially correlated is necessary. First it may open new perspectives to explain these losses and improve our understanding of the flaws of the hydrological model and of the EHUP. Furthermore, it would be very useful to help the operational forecasters to detect major events when their forecasts have to be particularly questioned.*

 *Furthermore, improving the regionalisation of the predictive distribution assessment, as proposed in Bourgin et al. (2015) and Bock et al. (2018) could help build more robust assessment of uncertainty quantification when forecasting high flows.*" |
| 297 | 37 | 13 | The sentence has been changed in "*As pointed out by McInerney* et al. *(2017), when alpha << y […]*" |
| 125 & 381 | 22 & 23 | | Fig. 12 & 13 have been merged |

(*) In *190803_Answers to revisions.pdf*. Before line 307 (blue): 1st (anonymous) referee ; after line 30 (green) : 2nd referee (Dr. Engeland)

| Line in the answer to referees' comments (*) | Page | Line | Changes made in the second submitted version |
|---|---|---|---|
| | in the first submitted version | | |
| 128 | 28 & 29 | | Fig. 17 & 18 have been moved to the supplementary materials |
| – | End | | A sentence was added to acknowledge the contributions of Dr. Engeland and of an anonymous referee. |

[revised manuscript text omitted]

---

## Referee Report (RR1)

**Second review for "A crash-testing framework for predictive uncertainty assessment when forecasting high flows in an extrapolation context"**

The revised manuscript is much improved over the original submission. In particle, the authors have addressed my main concerns by providing more details of EHUP, better organising the discussion and (slightly) reducing the number of figures. There are still some minor issues which I feel should be addressed, so I recommend minor revisions prior to publication.

**Comments**

**Abstract line 15:** "… the Box-Cox transformation with a parameter between 0.1 and 0.3 can be a reasonable choice for flood forecasting".

This is not a key finding of the paper – the impact of using values of lambda between 0.1 and 0.3 is not shown or discussed or even referred to in the main paper, but is hidden as figure 7 in the supplementary material. Therefore, this particular finding should not appear in the abstract.

**Section 1.1 title:** "The big one: dream or nightmare for the forecaster?"

This an interesting question, but it does not seem a relevant title for this section since the question is not discussed.

**Page 3, Lines 15-19:** You mention that Renard et al (2010) models each source of uncertainty, but they do this in a hydrological prediction context (not hydrological forecasting). They do not consider uncertainty in meteorological forcing. This should be mentioned.

**Page 3, Line 20:** "In particular, the ensemble approaches intend to account for meteorological forecast uncertainty."

The words "In particular, the ensemble …" do not work here. I suggest replacing with "Alternatively, the ensemble …". This will nicely contrast it to the work of Renard.

**Section 1.3:** In this study you do not consider uncertainty in meteorological forecasts. This should be stated explicitly and justified when defining the scope of this paper.

**Page 8, Lines 19-20**. "Here, this highest flow group contains the top 5% pairs ranked by forecasted values."

It is not clear if this is the top 5% of all data, or top 5% of training data.

**Page 8, Lines 23-24**: "The EHUP can be applied after a preliminary data transformation, and by adding a final step to back-transform the predictive distributions obtained in a transformed space."

What is the purpose of the data transformation in the EHUP? Please explain this.

**Page 9, Line 13:** "The Box-Cox transformation (Box and Cox, 1964) is a classic one-parameter transformation"

This statement is not really correct - the transformation you show has two parameters – lambda and a.

**Figure 2 caption:** "the log transformation (thick green straight line)."

This green line is not straight – it is curved.

**Page 10, Lines 1-10:** I still do not see the purpose of describing and providing an equation for the NQT since the NQT is not used in this study. You can explain that it is commonly used and provide reasons for why you don't use it, but no need to provide details about it.

**Section 2.2.3.:** You describe how G1, G2, G3 and D1, D2inf, D2sup and D3 are selected. But it is not clear how these relate to the top 5% of forecast data used for training the EHUP (Page 8, lines 9-11).

Are the data subsets based on all data, or only the top 5% of data? Or are the thresholds somehow chosen so that 5% of data is used?

**Page 13, Line 14:** "estimating 99 percentiles"

What do you mean by 99 percentiles? Do you mean the $99^{th}$ percentile (top 1%)? Or do you mean estimating percentiles between 0 and 100? If the latter, you can probably remove "99" from this sentence to make it clearer.

**Section 2.3.1:** What are good/bad values for each metric? You describe this for alpha-index, but not for other metrics (e.g. higher values for CRPSS are better, with 1 being perfect and 0 being same as climatology).

**Figure 5:** I still find this figure a bit confusing – I guess it is complicated figure since you are dealing with different periods for training (calibrating) the EHUP and calibrating the transformation parameters and evaluating the model. Here are some comments which may (or may not) help to make things clearer:

- In panel (a), it looks like the data transformation is calibrated after the EHUP is trained, which as far as I can tell is incorrect. For each set of transformation parameters, the EHUP is trained over the training period. Then based on performance over the calibration period, the set of calibrated parameters is chosen. So panel (a) is really about calibrating the transformation parameters.
- In panel (b), you are re-training the EHUP with the calibrated set of transformation parameters and additional data, and then evaluating performance using another set of data. Perhaps this can somehow be made clearer in the figure/caption.

**Figure 5 caption:** "The straight lines represent their use to assess …"

Do you mean the vertical lines?

**Figure 5 caption:** What are the grey dots?

**Page 18, Lines 9-10.** "They show quite different patterns on the set of catchments, highlighting bias or under-dispersion problems for some of them, as illustrated in Fig. 9."

This should be elaborated on. You provide details about this in the figure caption, but they should be provided in the text.

**Section 3.2.2 heading:** "Overall performance"

This section describes "overall performance" using the CRPSS metric. But it also describes accuracy and sharpness. The heading is misleading since it only mentions overall performance. Perhaps change to "Other performance metrics"

**Page 20, Line 11:** Reference to Supplementary Material. Please provide reference to specific figure numbers in supplementary material.

Same comment applies for other references to Supplementary Material.

**Section 4.2:** The results presented in this section are somewhat interesting, but I do not see how they address a specific aim of the paper. As such, they do not seem to add value to the paper and in my opinion are a distraction from the key findings of the paper.

---

## Referee Report (RR2)

**Third review for "A crash-testing framework for predictive uncertainty assessment when forecasting high flows in an extrapolation context"**

The authors have done well to address all of my previous comments, and I recommend this paper be published in HESS. I have a few comments which I believe will improve the readability of the paper.

**Main comment**

The term "non-parametric" has a specific meaning in the statistical literature, which implies that an underlying statistical distribution is **not** used. Examples of these include the NQT, as referred to in this paper. It is incorrect to refer to the Log and Box-Cox transformations used in this work as non-parametric transformations (even if they do not have any calibrated parameters). The authors should resolve this issue in order to avoid reader confusion.

**Specific comments**

**Page 2, lines 23-24:**

Change "In order to achieve *an* efficient crisis management and decision making"

to "In order to achieve efficient crisis management and decision making"

**Page 3, line 4:**

Change "The first step consists in identifying the different sources of uncertainty"

to "The first step consists of identifying the different sources of uncertainty"

**Page 3, lines 14-15:**

Change "(e.g., in their study on hydrological prediction, Renard et al. (2010) had not to consider the uncertainty in meteorological forecasts)"

to "(e.g., in their study on hydrological prediction, which did not consider uncertainty in meteorological forecasts)"

**Page 14, line 2:**

Change "Since there are only one parameter for the Box-Cox transformation"

to "Since there is only one parameter for the Box-Cox transformation"

**Page 21, lines 6-7:**

Change "In addition to reliability, we looked at other qualities of the probabilistic forecasts, namely the overall performance (measured by the CRPSS) and accuracy. We also checked their sharpness."

to "In addition to reliability, we looked at other qualities of the probabilistic forecasts, namely the overall performance (measured by the CRPSS), accuracy (measured by NSE) and sharpness (relative sharpness metric)."

---

## Author Response (AR2)

**A crash-testing framework for predictive uncertainty assessment when forecasting high flows in an extrapolation context: answers to the second revision**

Berthet, L., Bourgin F., Perrin, C., Viatgé, J., Marty, R. and Piotte O.

We are very grateful to both referees for their positive reviews and detailed comments and propositions. Here we provide some answers to their comments and the changes we made accordingly to finalize the article.

**Answer to the comments of the referee #1**

> *The revised manuscript is much improved over the original submission. In particle, the authors have addressed my main concerns by providing more details of EHUP, better organising the discussion and (slightly) reducing the number of figures. There are still some minor issues which I feel should be addressed, so I recommend minor revisions prior to publication.*
>
> *Abstract line 15: "… the Box-Cox transformation with a parameter between 0.1 and 0.3 can be a reasonable choice for flood forecasting".*
>
> *This is not a key finding of the paper – the impact of using values of lambda between 0.1 and 0.3 is not shown or discussed or even referred to in the main paper, but is hidden as figure 7 in the supplementary material. Therefore, this particular finding should not appear in the abstract.*

We agree that this is not a key finding of the paper. This mention has been removed from the abstract.

> *Section 1.1 title: "The big one: dream or nightmare for the forecaster?"*
>
> *This an interesting question, but it does not seem a relevant title for this section since the question is not discussed.*

We thank the reviewer for this comment. We agree that this title is not directly related to the entire content of this section: the 'fears' and concerns of operational forecasters are indeed not discussed from a 'psychological' point of view. However, this title was chosen for the introduction to introduce this first section since it explains "off the wall" why the issue discussed in the article is related to operational matters and considered as very important by operational forecasters. Since it is the introductory title, we prefer keeping it.

> *Page 3, Lines 15-19: You mention that Renard et al (2010) models each source of uncertainty, but they do this in a hydrological prediction context (not hydrological forecasting). They do not consider uncertainty in meteorological forcing. This should be mentioned.*

We agree with the reviewer. The main idea is that the methodology is based on a separate modelling of each **relevant** source of uncertainty. This point has been specified and we added that in their study context, they had not to consider the meteorological forcing uncertainty in the next paragraph.

> *Page 3, Line 20: "In particular, the ensemble approaches intend to account for meteorological forecast uncertainty."*
>
> *The words "In particular, the ensemble …" do not work here. I suggest replacing with "Alternatively, the ensemble …". This will nicely contrast it to the work of Renard.*

We agree and thank the reviewer for her or his suggestion: "*In particular, the ensemble*" do not work. The text has been changed in connection with the previous amendments.

> *Section 1.3: In this study you do not consider uncertainty in meteorological forecasts. This should be stated explicitly and justified when defining the scope of this paper.*

This point has been be mentioned explicitly and is briefly discussed it in the perspectives.

> *Page 8, Lines 19-20. "Here, this highest flow group contains the top 5% pairs ranked by forecasted values."*
>
> *It is not clear if this is the top 5% of all data, or top 5% of training data.*

The "5%" refers to the data used at every step. *E.g.*, during the training of the calibration step, we used the top 5% pairs on D1 + D2$_{Sup}$ data. A sentence has been added to better explain this.

> *Page 8, Lines 23-24: "The EHUP can be applied after a preliminary data transformation, and by adding a final step to back-transform the predictive distributions obtained in a transformed space."*
>
> *What is the purpose of the data transformation in the EHUP? Please explain this.*

We thank the reviewer for her or his careful reading. Indeed, the use of a preliminary data transformation is not clear at this point of the reading. The use of the data transformation can be justified by different grounds. For practical purposes, in order to improve the predictive performance. In a scientific perspective where the modelling is based on an assumption of the normality of the distributions, in order to better fulfil this assumption. Therefore it is difficult to give one purpose. We hope that this point becomes much clearer further.

> *Page 9, Line 13: "The Box-Cox transformation (Box and Cox, 1964) is a classic one-parameter transformation"*
>
> *This statement is not really correct - the transformation you show has two parameters – lambda and a.*

We thank the reviewer. Since the parameter *a* is chosen equal to 0, there is only one free parameter. This point has been explained in the revised version.

> *Figure 2 caption: "the log transformation (thick green straight line)."*
>
> *This green line is not straight – it is curved.*

Thank you! This has been corrected!

> *Page 10, Lines 1-10: I still do not see the purpose of describing and providing an equation for the NQT since the NQT is not used in this study. You can explain that it is commonly used and provide reasons for why you don't use it, but no need to provide details about it.*

Both reviewers mentioned this point. We agree that it is not necessary to provide the formula. However the NQT is commonly used and we reckon that it is necessary to explain why this transformation is not relevant in an extrapolation context and therefore not used in this study. This has been summarized.

*Section 2.2.3.: You describe how G1, G2, G3 and D1, D2inf, D2sup and D3 are selected. But it is not clear how these relate to the top 5% of forecast data used for training the EHUP (Page 8, lines 9-11).*

*Are the data subsets based on all data, or only the top 5% of data? Or are the thresholds somehow chosen so that 5% of data is used?*

The data subsets are based on all the data. The revised version indicates that the top 5% pairs are taken from the data sample used at every step of training.

*Page 13, Line 14: "estimating 99 percentiles"*

*What do you mean by 99 percentiles? Do you mean the 99th percentile (top 1%)? Or do you mean estimating percentiles between 0 and 100? If the latter, you can probably remove "99" from this sentence to make it clearer.*

We agree with the reviewer and thank her or him. The number "99" has been removed to make the sentence clearer.

*Section 2.3.1: What are good/bad values for each metric? You describe this for alpha-index, but not for other metrics (e.g. higher values for CRPSS are better, with 1 being perfect and 0 being same as climatology).*

We thank the reviewer for this suggestion. A few sentences have been added to give the best values for every main criterion. This suggestion is related to a remark by reviewer #2 (Dr. Engeland) who suggests to explain in the legends of many figures that the dashed lines represent the optimal values.

*Figure 5: I still find this figure a bit confusing – I guess it is complicated figure since you are dealing with different periods for training (calibrating) the EHUP and calibrating the transformation parameters and evaluating the model. Here are some comments which may (or may not) help to make things clearer:*

- *In panel (a), it looks like the data transformation is calibrated after the EHUP is trained, which as far as I can tell is incorrect. For each set of transformation parameters, the EHUP is trained over the training period. Then based on performance over the calibration period, the set of calibrated parameters is chosen. So panel (a) is really about calibrating the transformation parameters.*

- *In panel (b), you are re-training the EHUP with the calibrated set of transformation parameters and additional data, and then evaluating performance using another set of data. Perhaps this can somehow be made clearer in the figure/caption.*

This figure illustrates the whole methodology. It is difficult to summarize the latter in the legend. As suggested by the reviewer, the subtitle of the first panel has been changed as "*Step 1 – Calibration of the transformation parameters*" and the legend has been completed.

*Figure 5 caption: "The straight lines represent their use to assess …"*

*Do you mean the vertical lines?*

Actually, these are the horizontal lines. The word "*horizontal*" has been added in the caption.

*Figure 5 caption: What are the grey dots?*

The grey dots are the data pairs which are not used during the calibration step and then the evaluation step. Due to the fact that D1, D2 and D3 data are taken from events (from G1, G2 and G3), some time steps where the discharge is equal or higher than the minimum discharge of D1 are not selected (because not in a particular event).

*Page 18, Lines 9-10. "They show quite different patterns on the set of catchments, highlighting bias or under-dispersion problems for some of them, as illustrated in Fig. 9."*

*This should be elaborated on. You provide details about this in the figure caption, but they should be provided in the text.*

The details have been provided in the text: *"First, we conducted a visual inspection of the PIT diagrams, which convey an evaluation of the overall reliability of the probabilistic forecasts (examples in Fig. 9). In some cases, the forecasts are reliable (e.g., the Thérain River at Beauvais, 755 $km^2$: except if no transformation is used). Alternatively, these diagrams show quite different patterns on the set of catchments, highlighting bias (e.g., the Meu River at Montfort-sur-Meu, 477 $km^2$) or under-dispersion (e.g. the Aa River at Wizernes, 392 $km^2$, or the Sioulet River at Miremont (473 $km^2$) where the calibration on D2sup leads to log-sinh and Box-Cox transformations equivalent to no transformation, which turns out not to be relevant on the control data set where the log and the Box-Cox transformations are more reliable)."*

The legend of Fig. 9 has been shortened as *"Examples of PIT diagrams obtained on the control data set D3, with different transformations at four locations."*

*Section 3.2.2 heading: "Overall performance"*

*This section describes "overall performance" using the CRPSS metric. But it also describes accuracy and sharpness. The heading is misleading since it only mentions overall performance. Perhaps change to "Other performance metrics"*

We thank the reviewer for her or his suggestion. The text has been changed accordingly.

*Page 20, Line 11: Reference to Supplementary Material. Please provide reference to specific figure numbers in supplementary material.*

*Same comment applies for other references to Supplementary Material.*

We agree with the reviewer that it could ease the reading. We checked in the authors' guidelines (https://www.hydrology-and-earth-system-sciences.net/for_authors/manuscript_preparation.html). We are no sure that it is allowed to refer through numbering to figures in Supplementary Materials from the main text. We wrote Copernicus but received no answer. Therefore we did not change the referring. Moreover, we changed the numbering of figures in the Supplementary Materials as S1, S2, ...

*Section 4.2: The results presented in this section are somewhat interesting, but I do not see how they address a specific aim of the paper. As such, they do not seem to add value to the paper and in my opinion are a distraction from the key findings of the paper.*

Section 4.2 belongs to the Discussion and while we agree that the complementary results are not the key findings of the paper, we believe that they add value to the paper by providing a connexion to other existing studies and common practice in the hydrological community. We believe it is important that different approaches can be evaluated if we want to make progress as a community and this section is a contribution to that effort.

**Answer to the comments of the referee #2 (Dr. Engeland)**

> *I find that the paper has improved and can be published after some minor clarifications. The paper presents a topic of high interest, the presentation is now consistent, it will be easy for other researcher to repeat a similar experiment.*
>
> *Page **8**, Line 24-25*
>
> *Please add reference in this sentence:*
>
> *"In previous work, we used the log transformation because it ensures that no negative values are obtained when estimating the predictive uncertainty for low flows."*

A reference to Bourgin *et al.* (2014) has been added.

> *Page 10 Line 1-10*
>
> *It is not necessary to provide many details for a method you did not use. Either delete this part or summarize the content in maximum two sentences.*

Both reviewers mentioned this point. We agree that it is not necessary to provide the formula. However the NQT is commonly used and we reckon that it is necessary to explain why this transformation is not relevant in an extrapolation context and therefore not used in this study. This has been summarized.

> *Page 12 and Figure 4.*
>
> *For the event selection you write that "the event was kept […] if the peak value was higher than 50% of the highest discharge value of the time series.*
>
> *In Figure 4 that illustrates the event selection, it seems like the highest peak is around 110 m3/s, whereas some of the events with the lowest flood peaks are much smaller than 55 m3/s that is 50% of the highest peak. Could you explain?*

We thank very much the reviewer for his careful reading. It is a mistake in our description, the threshold we used to decide whether we keep the event or not is the median value of the time series and not 50 % of the highest peak. The text has been corrected.

> *Page 12 Line 10-15. The calibration and verification procedure could be better explained, see suggestion below.*
>
> *"We used a two-step procedure, as illustrated in Fig. 5. The first step was to calibrate the parametric transformations. For each transformation and for each parameter set, the empirical residuals were computed over D1 and D2sup where the EHUP was trained on the highest flow group of D1 (see section 2.1.3) and the calibration criterion was computed on D2sup. Indeed, the data transformations have almost no impact on the uncertainty estimation by EHUP on events of the same magnitude as those of the training subset. Therefore, the calibration subset has to encompass events of a larger magnitude (D2sup). The parameter set obtaining the best criterion value was selected. In the second step, the EHUP was trained on the highest flow group of D1, D2inf and D2sup combined and using the parameter set obtained during the calibration step. Then, the predictive uncertainty distribution was evaluated on the control data set D3. "*

We thank the reviewer very much for his proposition. The text has been changed accordingly.

*Figure 6, 10, 12, 13, 14,15: Maybe add one sentence in the figure captions what the dashed line represents.*

We agree. The sentence "*the optimal values are represented by the dashed lines.*" has been added.

*Why have you chosen the alpha-index to have a optimal value at 1 whereas the other scores have an optimal value at 100% ?*

Many dimensionless criteria whose best value is 1 are sometimes shown as %. However it is not very relevant here. The text and the figures have been changed in order to use 1 instead of 100% (except for the coverage ratio).

*Page 27, : Line 29: For clarity you could write: We used the Shapiro-Francia test where the zero hypothesis is that the data are normally distributed.*

We agree and thank the reviewer. The sentence has been modified accordingly.

**Supplementary Materials**

We made only two changes in the Supplementary Materials :

1. The new numbering of the figures (S1, S2, …) according to the authors' guidelines (https://www.hydrology-and-earth-system-sciences.net/for_authors/manuscript_preparation.html), see page 4 of this answer to comments.

2. The "rescaling" of Fig. S1, S2 and S3 for the dimensionless criteria (except for the coverage ratio which is still expressed as %), accordingly to the the suggestion of Dr. Engeland (see answer above to comment for Figure 13).

[revised manuscript text omitted]

---

## Author Response (AR3)

**A crash-testing framework for predictive uncertainty assessment when forecasting high flows in an extrapolation context**

Lionel Berthet, François Bourgin, Charles Perrin, Julie Viatgé, Renaud Marty and Olivier Piotte

We would like to thank the reviewers very much for their careful and positive reviews and their final corrections.

**Main comment**

*The term "non-parametric" has a specific meaning in the statistical literature, which implies that an underlying statistical distribution is not used. Examples of these include the NQT, as referred to in this paper. It is incorrect to refer to the Log and Box-Cox transformations used in this work as non- parametric transformations (even if they do not have any calibrated parameters). The authors should resolve this issue in order to avoid reader confusion.*

We agree that using "non-parametric" for both statistical approaches and scalar functions (such as the variable transformations) may be rather confusing for the reader. Therefore, we now keep this word only for the statistical approaches. The transformations are now described as with / without calibrated parameters, or as "calibrated" / "uncalibrated".

**Specific comments**

**Page 2, lines 23-24:**

*Change "In order to achieve an efficient crisis management and decision making"*
*to "In order to achieve efficient crisis management and decision making"*

We corrected this sentence accordingly.

**Page 3, line 4:**

*Change "The first step consists in identifying the different sources of uncertainty"*
*to "The first step consists of identifying the different sources of uncertainty"*

We corrected this sentence, as suggested.

**Page 3, lines 14-15:**

*Change "(e.g., in their study on hydrological prediction, Renard et al. (2010) had not to consider the uncertainty in meteorological forecasts)"*
*to "(e.g., in their study on hydrological prediction, which did not consider uncertainty in meteorological forecasts)"*

The previous sentence was slightly changed, because the methodology itself was not presented (for the first time by Renard *et al.*, 2010). Therefore the sentence was not change. Indeed the fact that the uncertainty in meteorological forecasts was not taken into account in this study is due to the fact that this study focuses on hydrological prediction (and not on forecasting).

**Page 14, line 2:**

*Change "Since there are only one parameter for the Box-Cox transformation"*
*to "Since there is only one parameter for the Box-Cox transformation"*

This sentence was not change because the full sentence is: "t*here are only one parameter [...] and two parameters [...]*". Both plural and singular seem correct (based on a short search on the Internet). However the singular option suggests an ellipsis interpretation (*there is one parameter [...] and **there are** two parameters [...]*") which may not be obvious for non native-English readers.

**Page 21, lines 6-7:**

*Change "In addition to reliability, we looked at other qualities of the probabilistic forecasts, namely the overall performance (measured by the CRPSS) and accuracy. We also checked their sharpness."*
*to "In addition to reliability, we looked at other qualities of the probabilistic forecasts, namely the overall performance (measured by the CRPSS), accuracy (measured by NSE) and sharpness (relative sharpness metric).*

We changed the sentences to add the criteria. We did not merged the two sentences in order to "separate" the forecasts qualities from the sharpness which is only a distribution property: a sharp forecasted distribution is not necessarily a good forecast *per se*.

**Additional changes**

Two minor changes were made in the sections "*Author contributions*" and "*Acknowledgements*".

[revised manuscript text omitted]